# Neural Neighborhood Search for Multi-agent Path Finding

**Zhongxia Yan, Cathy Wu**
MIT
{zxyan,cathywu}@mit.edu

## Abstract

Multi-agent path finding (MAPF) is the combinatorial problem of planning optimal collision-avoiding paths for multiple agents, with application to robotics, logistics, and transportation. Though many recent learning-based works have focused on large-scale combinatorial problems by guiding their decomposition into sequences of smaller subproblems, the combined spatiotemporal and time-restricted nature of MAPF poses a particular challenge for learning-based guidance of iterative approaches like large neighborhood search (LNS), which is already a state-of-the-art approach for MAPF even without learning. We address this challenge of neural-guided LNS for MAPF by designing an architecture which interleaves convolution and attention to efficiently represent MAPF subproblems, enabling practical guidance of LNS in benchmark settings. We demonstrate the speedup of our method over existing state-of-the-art LNS-based methods for MAPF as well as the robustness of our method to unseen settings. Our proposed method expands the horizon of effective deep learning-guided LNS methods into multi-path planning problems, and our proposed representation may be more broadly applicable for representing path-wise interactions.

## 1 Introduction

Due to recent advances in robotics and artificial intelligence, logistics operations in warehouses have become increasingly automated. As an example, hundreds to thousands of mobile robots work in warehouses to transport items to their required locations. Limited warehouse space, hardware, and time to fulfill orders provide immense challenges and opportunities to coordinate the tasks and movements of these robots to maximize throughput and minimize congestion. Given the criticality and massive throughput of modern supply chains, even small improvements in solution quality would translate to billions of packages delivered faster annually. However, optimal coordination of such robots in real time on a large scale, even in simplified discrete simulation, raises an intractable discrete optimization problem: the NP-hard multi-agent path finding (MAPF) problem models the coordination of a set of agents, each with a start location and goal location, and requests a collision-free plan to move all agents from their start locations to their goal locations which optimizes the aggregate delay of the agents. While MAPF naturally considers mobile agents like robots in a warehouse or vehicles in a transportation system, other related multi-path planning problems may include routing of circuits (Cheng et al., 2022) or pipes.

Large-neighborhood search (LNS) is a general approach for decomposing large-scale optimization (Shaw, 1998) into a sequence of *subproblems*. LNS has been demonstrated to obtain state-of-the-art solution qualities for MAPF (Li et al., 2021a; 2022). While studies have shown the effectiveness of deep learning for guiding LNS in other combinatorial problems like integer programming (IP) (Wu et al., 2021; Huang et al., 2023) and vehicle routing problems (VRP) (Li et al., 2021b), such an approach has remained elusive for MAPF.

In this work, we identify the unique two-fold challenge of leveraging deep neural networks to guide LNS for MAPF. 1) Unlike many combinatorial problems like IP and VRP with graphical structure among variables or entities (Vinyals et al., 2015; Khalil et al., 2017; Paulus et al., 2022; Labassi et al., 2022; Scavuzzo et al., 2022), agent-to-agent interactions in multi-path planning problems *also* correspond to moments in space-time, where locality relationships must be aptly represented. 2) Due

to the iterative nature of LNS and the short runtime of multi-path planners, each neural decision for guiding LNS must be made in very limited time (typically less than 0.05s). Essentially, challenge 1 prescribes cumbersome convolutional architectures for representing MAPF subproblems, while challenge 2 demands very low inference time. For example, to sidestep these challenges, Huang et al. (2022) applies a linear model with hand-designed features while suggesting that a graph-convolution-based solution is impractical due to the inference time.

Addressing these challenges in our work, our main contributions are:

1. the first deep architecture designed to enhance LNS for MAPF by selecting agent subsets,

2. runtime speedup over state-of-the-art LNS baselines for obtaining a given solution quality,

3. empirical analyses of the generalization of our architectures to unseen settings.

Full code, models, and instructions can be found on GitHub upon publication.

## 2 MULTI-AGENT PATH FINDING

A multi-agent path finding problem is denoted by $P = (G, s_A, g_A)$, where agents $A = \{1, \ldots, |A|\}$ may move within a undirected graph $G$ with vertices and edges $(V, E)$. Each agent $a \in A$ has a start $s_a \in V$ and goal $g_a \in V$. Note that for conciseness, we use vectorized subscript $s_A = \{s_a \mid a \in A\}$. An agent transition from vertex $v \in V$ to $v' \in V$ is valid if $(v, v') \in E$. For all $v \in V$, staying is a valid move, *i.e.* $(v, v) \in E$. The path $p_a$ of agent $a$ is a sequence of vertices with *length* $\tau(p_a) := |p_a| - 1$ such that all transitions $(p_a[t], p_a[t + 1])$ with $0 \leq t \leq \tau(p_a)$ are valid and the path starts at $s_a$ and end at $g_a$. The shortest distance $d(v, v')$ between two vertices $v, v' \in V$ is the length of a shortest possible path $p(v, v')$ between $v$ and $v'$. The cost of an agent's path $p_a$ is $c(p_a) = \tau(p_a) - d(s_a, g_a)$ which is defined to be the *delay* that the agent suffers. A collision occurs if two agents $a$ and $a'$ occupy the same vertex or edge at the same timestep, *i.e.* $p_a[t] = p_{a'}[t]$ or $(p_a[t], p_a[t+1]) = (p_{a'}[t+1], p_{a'}[t])$ and time $0 \leq t \leq \max(\tau(p_a), \tau(p_{a'}))$, which is the common, stay-at-goal formulation (Stern et al., 2019). A solution $S = \{p_a | a \in A\}$ is a set of agent paths and is feasible if no conflicts exist. We consider a MAPF objective of total delay minimization: $\min_S c(S) = \sum_{p_a \in S} c(p_a)$ such that $S$ is feasible.

## 3 RELATED WORK

**Multi-agent Path Finding**   Nearly all methods for solving MAPF rely heavily on running single-agent path planner such as A* search (Hart et al., 1968) or SIPP (Phillips and Likhachev, 2011) multiple times, while holding paths of some set of other agents as constraints. A simple MAPF algorithm is prioritized planning (PP) (Erdmann and Lozano-Perez, 1987), also known as cooperative A* (Silver, 2005), which plans one agent path at a time in random agent order while avoiding collisions with all previously planned agent paths and obstacles. Conflict-based Search (CBS) (Sharon et al., 2015) is a seminal algorithm which relies on backtracking tree-search to recursively resolve pairs of agent collisions. As part of our algorithm, we leverage a more scalable extension of CBS called Priority-based Search (PBS) (Ma et al., 2019), which heuristically reduces the number of pairwise conflicts to resolve but is suboptimal and incomplete.

**Constructive vs Iterative Learning-based Methods**   Learning-based methods for optimizing combinatorial problems typically can be classified as constructive or iterative Kwon et al. (2020). Constructive methods such as Vinyals et al. (2015); Kool et al. (2019); Kwon et al. (2020) are trained to autoregressively generate one or more feasible solutions. On the other hand, iterative methods, including LNS-based methods in integer programming, ILP, and VRPs (Wu et al., 2021; Huang et al., 2023; Li et al., 2021b), start from an initial (often poor but feasible) solution and iteratively modify the solution to improve the solution quality. Constructive methods are often very fast in practice, but could yield lesser solution qualities than iterative methods.

**Large Neighborhood Search for MAPF**   MAPF-LNS (Li et al., 2021a) starts from an initially feasible solution then iteratively selects heuristically constructed subsets of agent paths to reoptimize while holding the rest constant. MAPF-ML-LNS (Huang et al., 2022) learns a linear model to rank

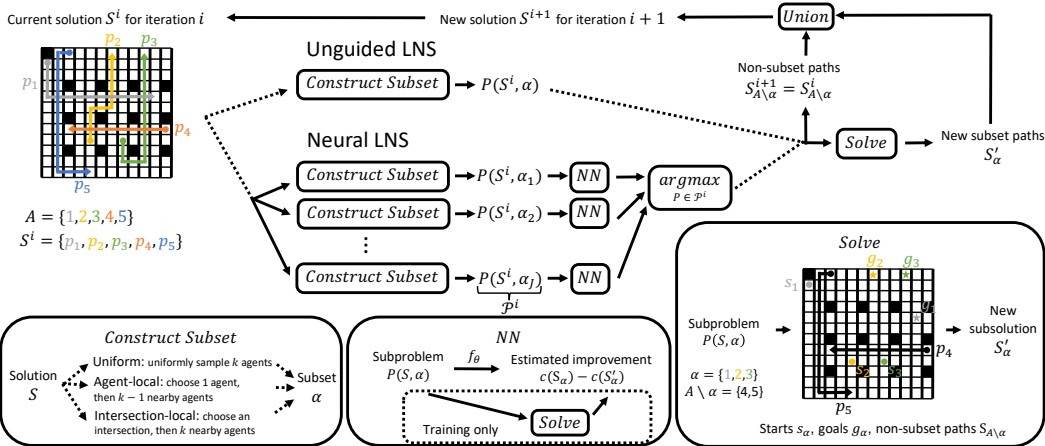

Figure 1: **Our neural LNS framework for MAPF** (counterclockwise from top left). Dotted (solid) arrows indicate that one (all) branch(es) must be taken. At each LNS iteration $i$, we have current feasible solution $S^i$. In unguided LNS, a subset $\alpha$ is constructed with either Uniform, Agent-local, or Intersection-local heuristics. In neural LNS, $J$ subsets are constructed, their corresponding subproblems are evaluated by the neural network, and the subproblem $P(S^i, \alpha)$ with the best estimated improvement is selected. In both cases, the subproblem $P(S^i, \alpha)$ is then solved to obtain new subsolution $S'_\alpha$, which forms part of the solution $S^{i+1}$ at the next iteration $i+1$ if its solution quality improves upon $S^i_\alpha$.

the potential improvements of these subsets and prioritize which subset to reoptimize. MAPF-LNS2 (Li et al., 2022) considers infeasible starting and intermediate solutions. These LNS-based methods obtain state-of-the-art solution qualities for problems where a feasible solution can be found, while orthogonal recent work like LaCAM* (Okumura, 2023) instead focus on state-of-the-art feasibility in very-dense settings. Given the orthogonality of feasibility and solution quality, we choose to extend MAPF-LNS rather than MAPF-LNS2. Unlike MAPF-ML-LNS, we efficiently represent the spatiotemporal problem structure with deep neural network rather than handcrafting features.

**Other Learning-based Approaches for MAPF** Huang et al. (2021) leverages a linear model with handcrafted features to guide CBS in problems with at most 200 agents. Damani et al. (2021) applies multi-agent reinforcement learning to decentrally coordinate agents based on local neighborhoods encoded by convolutional neural networks (CNNs), rewarding each agent when it reaches a goal. However, while this approach is scalable, its solution qualities are inferior to PBS and MAPF-LNS due to limitations of decentralization. Ren et al. (2021) applies 2D CNN to classify the effectiveness of solvers in small MAPF problems with less than 100 agents.

## 4 NEURAL LARGE NEIGHBORHOOD SEARCH FOR MAPF

We illustrate our overall framework in Figure 1. Given the MAPF problem $P$, let $S^0$ be a feasible initial solution. For each improvement iteration $0 \le i < I$, let $S^i$ be the current (feasible) solution.

In unguided LNS, a subset of agents $\alpha \subset A$ of size $k$ is constructed with one of three subset-construction heuristics described in Li et al. (2021a):

1. Uniform: uniformly randomly sample $k$ agents to construct $\alpha \subset A$.

2. Agent-local: sample one agent with nonzero delay which has not been recently sampled, then perform a random walk to sample $k-1$ nearby agents in space and time.

3. Intersection-local: sample any vertex of degree $\ge 3$, then perform a random walk to sample $k$ agents near the intersection.

Given $S^i$ and agent subset $\alpha$ to reoptimize, we define the corresponding *subproblem* $P(S^i, \alpha) = (G, s_\alpha, g_\alpha, S^i_{A \setminus \alpha})$ to be a *constrained* MAPF problem with $k$ agents. The current paths $S^i_\alpha$ for the

$\alpha$ agents are discarded. The paths of the $|A| - k$ non-subset agents $A \setminus \alpha$ are held constant as spatiotemporal obstacles, essentially changing the static graph $G$ to be a time-varying graph. The subproblem can be solved by a MAPF solver, *e.g.* Priority-based Search (PBS), to obtain a new subsolution $S'_\alpha = \text{Solver}(P(S^i, \alpha))$. The solution at time $i + 1$ is then $S^{i+1} \leftarrow S'_\alpha \cup S^i_{A \setminus \alpha}$ if the *improvement* $\delta_{\text{Solver}}(S^i, \alpha) = c(S^i_\alpha) - c(S'_\alpha)$ is positive, otherwise $S^{i+1} \leftarrow S^i$.

In neural LNS, we heuristically construct $J$ agent subsets $\{\alpha_1, \ldots, \alpha_J\}$ corresponding to a set of subproblems $\mathcal{P}^i = \{P(S^i, \alpha_j) | 1 \leq j \leq J\}$. Each subproblem $P(S^i, \alpha_j)$ is evaluated with a trained neural network $f_\theta(P(S^i, \alpha_j))$ whose output is a score approximating the quality of the true solver improvement $\delta_{\text{Solver}}(P(S^i, \alpha_j))$. The top-scored subproblem $P(S^i, \alpha) = \text{argmax}_{P \in \mathcal{P}^i} f_\theta(P)$ is selected, solved, and incorporated into the full solution. As discussed in Section 3, neural LNS is similar in structure to MAPF-LNS and MAPF-ML-LNS.

## 4.1 CHALLENGES OF GUIDING LNS

The key challenge of designing a deep neural network $f_\theta$ for guiding LNS for MAPF is to design a suitable representation for subproblem $P(S^i, \alpha) = (G, s_\alpha, g_\alpha, S^i_{A \setminus \alpha})$. Previous non-LNS-based MAPF works (Ren et al., 2021) use 2D CNN to encode 2D spatial tensors representing obstacles in $G$ and starts $s_\alpha$ and goals $g_\alpha$ of agents. However, due to the LNS context, we must also represent *hundreds* of paths $S^i_{A \setminus \alpha}$ of non-subset agents $A \setminus \alpha$ as hard occupancy constraints to the subproblem. To capture these dense spatiotemporal relationships, we utilize 3D convolutions in our architectures. In contrast, MAPF-ML-LNS (Huang et al., 2022) designs linear summary features for $S^i_{A \setminus \alpha}$ (see Appendix A.4); meanwhile, a similar work for VRP (Li et al., 2021b) does not need to capture the non-subset agents $A \setminus \alpha$ at all due to the lack of space-time interactions between $\alpha$ and $A \setminus \alpha$.

## 4.2 NAIVE PER-SUBSET ARCHITECTURE

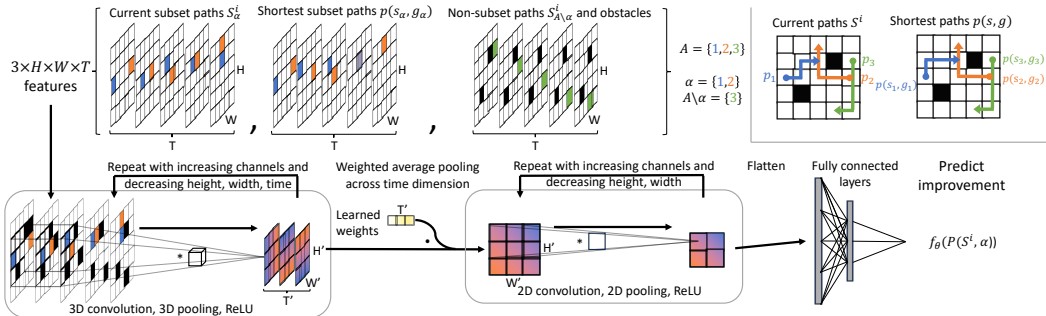

Figure 2: **Per-Subset architecture** consisting of featurized paths and obstacles, 3D convolutions, 2D convolutions, and finally a fully connected network. The current paths $S^i$ and shortest paths $p(s, g)$ for all agents are illustrated for reference (top right).

Intuitively, we propose to featurize $S^i_{A \setminus \alpha}$ as a 3D, one-hot encoded $H \times W \times T$ *obstacle* tensor, where $T$ is a time cutoff chosen roughly as the average path length of the agents. Specifically, for each agent $a$ at time $0 \leq t \leq \min(\tau(p_a), T - 1)$ and $(y, x) = p_a[t]$, we set entry $(y, x, t)$ in the tensor. Similarly, for every non-moving obstacle at position $(y, x)$, we set entry $(y, x, t)$ in the obstacle tensor for all $0 \leq t < T$. To maximize spatiotemporal alignment with the obstacle tensor, we additionally one-hot encode the current subset paths $S^i_\alpha$ into a 3D tensor, providing information to quantify $\delta(P(S^i, \alpha)) = c(S^i_\alpha) - \text{Solver}(P(S^i, \alpha))$. Finally, we one-hot encode the shortest subset paths $p(s_\alpha, g_\alpha) = \{p(s_a, g_a) | a \in \alpha\}$ into a third 3D tensor, providing information on potential gains of running the solver. Despite these features, the truncation of long paths and the one-hot encoding of subset paths unavoidably lose some information.

We stack the three tensors into a $3 \times W \times H \times T$ tensor as input into the neural network. Our convolutional architecture, shown in Figure 2, applies a sequence of 3D convolution blocks, aggregates along the temporal dimension, applies a sequence of 2D convolution blocks, flattens, then applies a fully connected network. We denote this architecture as the Per-Subset architecture.

### 4.3 MULTI-SUBSET ARCHITECTURE

Our core design principles for an effective architecture for guiding LNS are 1) to permit *trajectory-level* information to flow between any two agents with intersecting or adjacent paths, and 2) amortize computation across multiple subproblems. While 3D convolution alone efficiently encodes spatiotemporal agent-agent and agent-obstacle interactions at any point along agent paths, convolution is inherently a local operation and cannot efficiently capture long-range *trajectory-level* interactions between the agents, motivating an attention-based mechanism. We illustrate our desired inter-agent and intra-agent interaction graph in the top left diagram in Figure 3. Additionally, as the Per-Subset architecture encodes a single subproblem, it requires additional batching across $J$ subproblems for neural LNS. We hypothesize that a new architecture representing trajectory-level interactions between all agents would permit significant shared computation among the $J$ subproblems $P(S^i, \alpha_j)$, reducing the total computation for large $J$. As the representation can be disaggregated into individual agent trajectories, we can regroup individual agent representations efficiently into arbitrary number of subsets as required by LNS.

We propose an *intra-path* attention mechanism which, in alternation with 3D convolutions, achieves the desired agent interaction graph. Intra-path attention enables information flow along each agent's current path; 3D convolution provides for information flow between nearby space-time. Essentially, any two agents are two-hop neighbors in this representation if their paths both interact with some common agent at any point. For efficiency, we implement our intra-path through a series of "gather" and "scatter" tensor operations surrounding a Transformer self-attention (Vaswani et al., 2017).

We denote this as the Multi-Subset architecture. The input is three $H \times W \times T$ tensors corresponding to all agents' paths $S^i$, all agents' shortest paths $p(s_A, g_A)$, and static obstacles. A series of convolution-attention blocks is shared across all subsets. Given a particular subset of agents $\alpha_j$, we extract the first temporal tensor for each agent from the last intra-path attention layer's output, then perform additional layers of Transformer self-attention to predict the score $P(S^i, \alpha_j)$. Therefore, heavy 3D convolution and intra-path operations are amortized across all subsets.

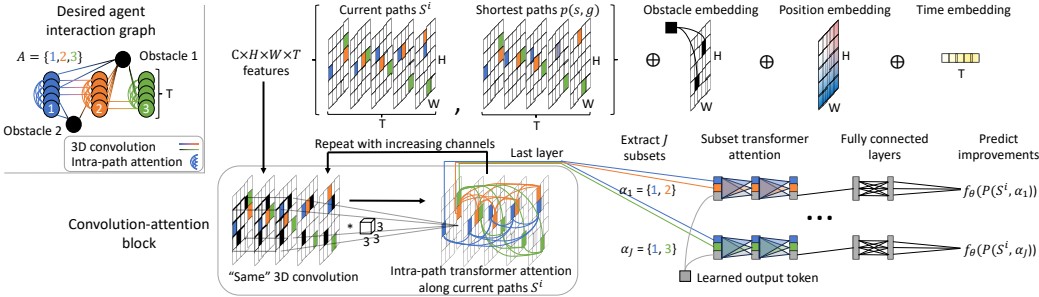

Figure 3: **Multi-Subset architecture** is designed to encode both inter-agent interactions between agent paths and obstacles as well as intra-path interactions for obtaining *trajectory-level* information (top left). 3D convolutions encode local interactions in space-time, while intra-path attention generates trajectory-level information across an agent's entire path. Convolution-attention blocks shared across all subsets. The current paths $S^i$ and shortest paths $p(s, g)$ are the same as those in Figure 2.

### 4.4 PAIRWISE CLASSIFICATION LOSS FUNCTION

Since the Multi-Subset architecture encodes $J$ subsets corresponding to the same $S^i$ at the same time, we design a pairwise classification loss: for every pair of subsets $\alpha_j$ and $\alpha_{j'}$, let the difference in predicted scores be $h_\theta(S^i, \alpha_j, \alpha_{j'}) = f_\theta(P(S^i, \alpha_j)) - f_\theta(P(S^i, \alpha_{j'}))$. A hinge loss encourages $h_\theta(S^i, \alpha_j, \alpha_{j'})$ to be positive when $\delta_{\text{Solver}}(P(S^i, \alpha_j)) > \delta_{\text{Solver}}(P(S^i, \alpha_{j'}))$ and negative vice versa; the loss is 0 when $\delta_{\text{Solver}}(P(S^i, \alpha_j)) = \delta_{\text{Solver}}(P(S^i, \alpha_{j'}))$. In summary, this loss encourages the ordering of predicted scores to match the ordering of ground-truth improvements:

$$\ell(h_\theta) = \begin{cases} \max(0, 1 - h_\theta), & \text{if } \delta_{\text{Solver}}(P(S^i, \alpha_j)) > \delta_{\text{Solver}}(P(S^i, \alpha_{j'})), \\ \max(0, 1 + h_\theta), & \text{if } \delta_{\text{Solver}}(P(S^i, \alpha_j)) < \delta_{\text{Solver}}(P(S^i, \alpha_{j'})), \\ 0, & \text{if } \delta_{\text{Solver}}(P(S^i, \alpha_j)) = \delta_{\text{Solver}}(P(S^i, \alpha_{j'})). \end{cases} \quad (1)$$

## 5    EXPERIMENTAL SETUP

We briefly discuss experimental setup in the main text, deferring full details to Appendix A.1.

**Baselines.** To analyze the performance of our proposed Multi-Subset architecture, we implement the Unguided baseline similarly as MAPF-LNS (Li et al., 2021a), the Linear baseline with features from MAPF-ML-LNS (Huang et al., 2022), and the Per-Subset architecture baseline described above.

**Floor Maps.** We demonstrate our methods on floor maps, which define the undirected graph $G$, from the MAPF benchmark suite Stern et al. (2019). As illustrated in Figure 4, we use the five largest floor maps studied by Li et al. (2021a): empty-32-32, random-32-32-10, warehouse-10-20-10-2-1, ost003d, and den520d. For readability, we annotate each floor map by their size: empty (32x32), random (32x32), warehouse (161x63), ost003d (194x194), and den520d (256x257). Starts and goals are each sampled uniformly without replacement from the set of non-obstacle vertices.

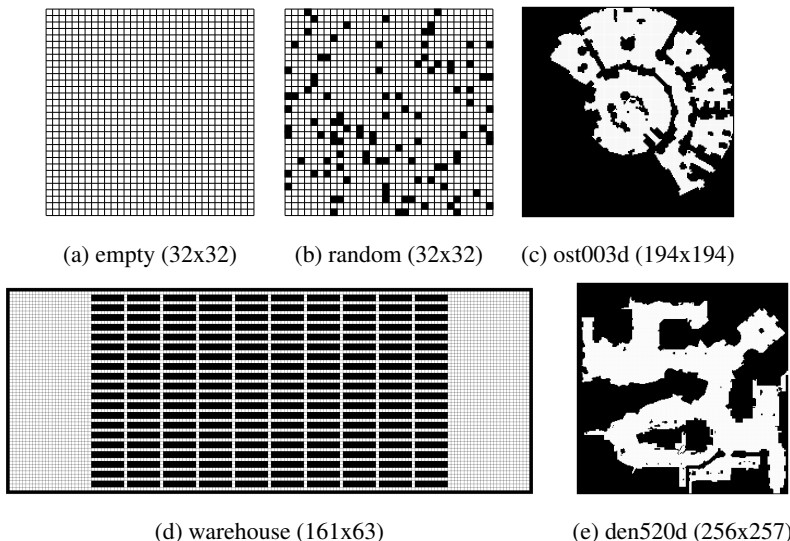

(a) empty (32x32)          (b) random (32x32)          (c) ost003d (194x194)

(d) warehouse (161x63)                    (e) den520d (256x257)

Figure 4: **Warehouse floor maps** from MAPF benchmark suite (Stern et al., 2019).

**Initial Solution.** The initialization algorithm does not typically affect the effectiveness of LNS-based approaches (Li et al., 2021a), as long a feasible solution can be reliably found. Prioritized planning (PP) (Erdmann and Lozano-Perez, 1987) typically obtains better initial solution quality than Parallel Push and Swap (PPS) Sajid et al. (2012), but the latter more often finds a feasible solutions (Li et al., 2021a). Thus, we use PP if it can reliably find a feasible solution for a particular setting, and otherwise use PPS.

**Solver, Problem Size, Subset Construction, Subset Size.** As there are many potential combinations of MAPF solvers (*e.g.* Priority-based Search (PBS) vs PP with a random agent order), subset construction heuristics (Uniform vs Agent-local vs Intersection-local), and subset size $k$, we perform parameter sweeps detailed in Appendix A.2 to first identify the strongest possible configuration for the Unguided baseline. We find that PP typically has worse solution quality to time tradeoff than the PBS. Considering the ranges of problem sizes $|A|$ studied by MAPF-LNS (Li et al., 2021a), we find that PBS offers better quality to time tradeoffs than PP in 2/5 of the settings for empty-32-32, 5/5 settings for random-32-32-10, and 4/5 settings for each of the other three floor maps. Our neural experiments focus on these settings where the PBS solver is stronger, as the determinism of PBS (PP is stochastic) can be more consistently predicted and the slower runtime of PBS benefits more from neural guidance. For each floor map, we focus on the largest problem size $|A|$ where PBS is the stronger solver; typically Agent-local subsets with $k = 25$ is the strongest. In very-dense highly-constrained settings, *e.g.* empty-32-32 with $|A| \geq 400$ agents or den520d with $|A| = 900$, we find that PP with very small $k = 5$ often is the best unguided LNS configuration. While our method is likely applicable to these cases, we acknowledge that the overhead of the neural network may be significant if the runtime of PP $k = 5$ were very short. Unlike Huang et al. (2022), we do

not allow our model to select subsets with different construction heuristics or size since the runtime of the solver on those subsets could be substantially different. Future work could additionally predict the runtime of the solver on the subset in addition to the improvement, trading-off runtime and improvement in order to select the best subset.

**Data Collection.** To collect training data from a random seed, we execute LNS for 25 to 100 improvement iterations: for each iteration $i$, we *enumerate* the solver on $J = 100$ agent subsets and record all triplets $(S^i, \alpha_j, \delta_{\text{PBS}}(P(S^i, \alpha_j)))$ as training data. To proceed to iteration $i + 1$, we select the subset with the best improvements $\delta_{\text{PBS}}$, and move to iteration $i + 1$. We use up to 9000 different initial random seeds. Data collection takes around 10 to 50 hours on 48 Intel Xeon Platinum 8260 CPU cores, depending on floor map size.

**Architectural Parameters.** For the Per-Subset architecture, we use two 3D convolutional blocks with 32 and 64 channels, respectively, followed by two 2D convolutional blocks with 128 channels each. For the Multi-Subset architecture, we use eight convolution-attention blocks with 16 to 128 channels, shared across all subsets, followed by three Transformer multi-head attention blocks with 128 features for each subset. We provide additional architectural ablations in Appendix A.6. The Linear baseline utilizes handcrafted features described in Huang et al. (2022) and Appendix A.4.

**Training.** Multi-Subset and Linear are trained with the pairwise classification loss in Section 4.4, while Per-Subset utilizes a clipped regression loss described in Appendix A.1. Training takes less than 24 hours on a single NVIDIA V100 GPU. Hyperparameters of all models are manually tuned on one experimental setting, then replicated across all other experimental settings.

**Validation and Test.** We validate and test our trained models on different random seeds. The final test results are reported as 95% confidence intervals across roughly 200 previously unseen seed values where feasible initial solutions could be found. For all settings, we assume a planning time limit of $T_{\text{limit}} = 60\text{s}$ unless otherwise stated. For absolute comparisons, Appendix A.7 and A.8 respectively compares sums of delays $c(S)$ with respect to runtime and improvement iterations.

**Model Overhead Reduction.** As MAPF solvers are fast and any model inference time adds to the overall stepwise runtime, we devise techniques to further reduce inference overhead. If necessary, we pre-apply 2D spatial pooling along both $W$ and $H$ for all agent locations and obstacles to reduce the floor map size. We similarly pre-apply temporal pooling if necessary to minimize redundant information. All guided methods utilize a single NVIDIA V100 GPU with FP16 precision for acceleration. Interestingly, we find that the CPU overhead of Linear is significant, so we instead compute features for Linear on the GPU to reduce overhead.

**Metrics.** For a given seed, denote the solution cost achieved by running Unguided for 600 seconds as $c_{\text{min}}$. We define the gap as $g(S) = \frac{c(S) - c_{\text{min}}}{c(S^0) - c_{\text{min}}} 100\%$, which better gauges the suboptimality of a solution. Given a solution quality $g$, we define the *speedup* of method $X$ over method $Y$ as the time for method $Y$ to attain $g$ divided by the time of method $X$ to attain $g$. Following Li et al. (2021a), we define the *area under curve (AUC)* of a method as $\text{AUC} = \int_0^{T_{\text{limit}}} g(S^{i(t)}) \, \mathrm{d}t$, which takes into account not only the final solution cost but also the rate of decrease. However, we instead report the more interpretable $\text{AUC}/T_{\text{limit}}$, the *average gap*. We define the Win / Loss metric as the number of seeds where a method's average gap is lower / higher than that of Unguided on the same seed.

## 6 EXPERIMENTAL RESULTS

### 6.1 PERFORMANCES ON ALL SETTINGS

In Table 1, we compare the performance and overheads of all methods under all floor maps. Multi-Subset strongly outperforms all other methods in empty (32x32), warehouse (161x63), and ost003d (194x194), and weakly outperforms Linear in random (32x32) and den520d (256x257). Multi-Subset's overhead is 2x less than Per-Subset for the empty and random settings, which do not pre-apply pooling. Surprisingly, we see that pre-applying 2x temporal, 4x spatiotemporal, and 4x spatiotemporal pooling for warehouse, ost003d, and den520d still allows Multi-Subset to maintain best predictivity while keeping the model overhead low. On the other hand, when pre-applying similar amount of pooling for the Per-Subset architecture for ost003d and den520d, the model overhead even exceeds the solver runtime.

Table 1: **Performance and overhead of all methods**. We report the the average solver time per step for the Unguided overhead. All other overheads are additionally incurred by model inference. Gaps and Win / Loss account for the model overhead already. Lower gap is better; higher Win is better.

| Setting | Metric | Unguided | Linear | Per-Subset | Multi-Subset |
|---------|--------|----------|--------|------------|--------------|
| **empty (32x32)** $\|A\| = 350$ Uniform $k = 50$ | Average Gap (%) | 42 ± 1 | 39 ± 1 | 46 ± 1 | **31 ± 1** |
| | Win / Loss | 0 / 0 | 136 / 50 | 57 / 129 | **178 / 8** |
| | Final Gap (%) | 19 ± 1 | 18 ± 0.8 | 22 ± 1 | **13 ± 0.7** |
| | Overhead (s) | 0.1 | +0.002 | +0.03 | +0.013 |
| **random (32x32)** $\|A\| = 250$ Agent-local $k = 25$ | Average Gap (%) | 5.3 ± 0.2 | 5.4 ± 0.3 | 6.2 ± 0.2 | **5 ± 0.2** |
| | Win / Loss | 0 / 0 | 98 / 100 | 36 / 162 | **116 / 82** |
| | Final Gap (%) | **0.51 ± 0.03** | 0.7 ± 0.05 | 0.72 ± 0.05 | 0.69 ± 0.04 |
| | Overhead (s) | 0.079 | +0.0018 | +0.03 | +0.012 |
| **warehouse (161x63)** $\|A\| = 300$ Agent-local $k = 25$ | Average Gap (%) | 14 ± 0.6 | 15 ± 0.7 | 20 ± 0.8 | **12 ± 0.6** |
| | Win / Loss | 0 / 0 | 82 / 103 | 6 / 179 | **155 / 30** |
| | Final Gap (%) | 1.7 ± 0.2 | 2.5 ± 0.3 | 2.5 ± 0.3 | **1.6 ± 0.2** |
| | Overhead (s) | 0.21 | +0.0025 | +0.05 | +0.025 |
| **ost003d (194x194)** $\|A\| = 400$ Agent-local $k = 10$ | Average Gap (%) | 43 ± 1 | 36 ± 1 | 35 ± 0.9 | **22 ± 1** |
| | Win / Loss | 0 / 0 | 180 / 20 | 188 / 12 | **200 / 0** |
| | Final Gap (%) | 24 ± 1 | 15 ± 1 | 16 ± 0.8 | **6.5 ± 0.7** |
| | Overhead (s) | 0.063 | +0.002 | +0.09 | +0.013 |
| **den520d (256x257)** $\|A\| = 800$ Agent-local $k = 25$ | Average Gap (%) | 45 ± 0.7 | 30 ± 0.7 | 48 ± 0.7 | **29 ± 0.7** |
| | Win / Loss | 0 / 0 | **200 / 0** | 47 / 153 | **200 / 0** |
| | Final Gap (%) | 22 ± 0.7 | 8.8 ± 0.4 | 22 ± 0.6 | **8.2 ± 0.5** |
| | Overhead (s) | 0.15 | +0.006 | +0.19 | +0.025 |

## 6.2 TIME VS SOLUTION QUALITY FOR EMPTY (32X32) AND DEN520D (256X257)

In Figure 5, we illustrate representative time vs gap tradeoff of all methods on the smallest and largest floor maps. Similar plots for the remaining floor maps are found in Appendix A.3. We observe here that Multi-Subset can offer a 1.5-4x speedup compared to Unguided, though Linear also offers substantial speedup on den520d (256x257) in particular. The model overhead is negligible for Linear, still noticeable for Multi-Subset, but large for Per-Subset. Optionally excluding the overhead allows us to independently judge the predictivity of the model's representation.

## 6.3 GENERALIZATION TO UNSEEN SETTINGS

In Table 2, we test the robustness of each method to unseen settings during training by examining the zero-shot transfer performance of two models from Table 1: the model trained on empty (32x32) with $\|A\| = 350$ Uniform $k = 50$ subsets and the model trained on random (32x32) with $\|A\| = 250$ Agent-local $k = 25$ subsets. The target settings are empty (32x32) settings with different numbers of agents or subset construction heuristics as determined by the parameter sweep in Appendix A.2. We find that the multi-subset architecture transfers particularly well, even better than Linear, to the unseen settings. This is particularly surprising for transferring from the random (32x32) source task, where the performance gap between the Linear and Multi-Subset models is small. Unfortunately, we could not transfer from empty (32x32) to random (32x32) because the obstacle embedding is untrained in the former setting due to lack of obstacles. Similarly, we could not transfer between floor maps of different size due to differing spatial and temporal embeddings.

## 7 CONCLUSIONS

In this work, we design a neural architecture with potential for effectively guiding LNS for MAPF, efficiently representing complex spatiotemporal and agent-to-agent interactions without hand-designed features. While we acknowledge that fair computational resource comparisons are

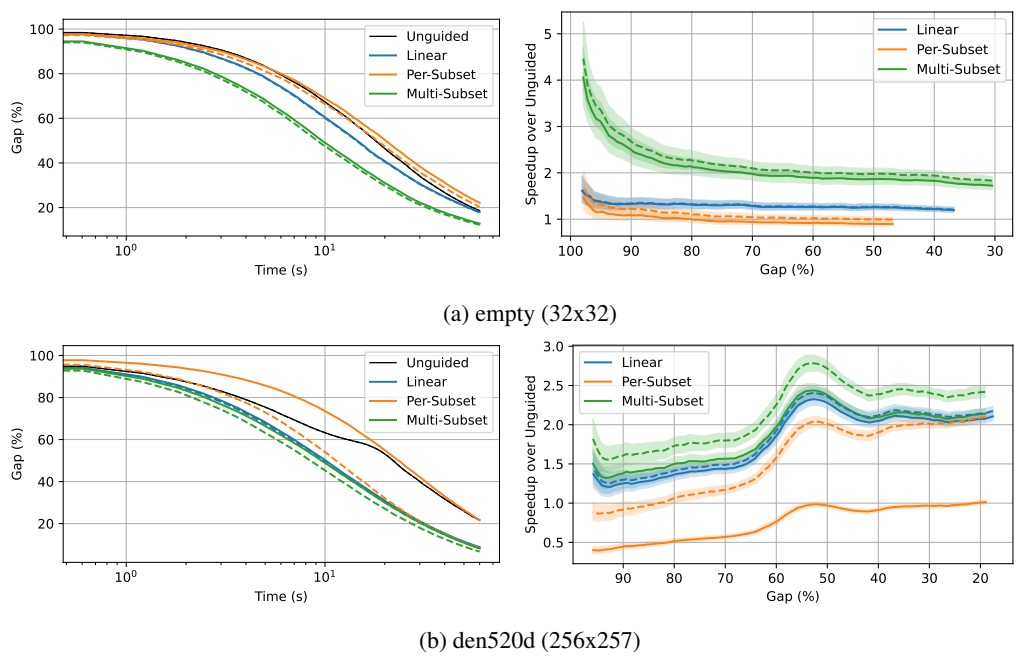

(a) empty (32x32)

(b) den520d (256x257)

Figure 5: **Runtime vs solution quality** including (solid) and excluding (dashed) model overhead in the runtime. (left) Lower gap is better. (right) Higher speedup over Unguided is better.

Table 2: **Zero-shot generalization performance.** As an example, "empty, 350, A25" denotes the empty (32x32) setting with $|A| = 350$ and Agent-local $k = 25$ subsets. Average gap (%) are reported for all settings. As the solver runtime is significantly faster for easier target settings, affecting the relative model overhead, we exclude the model overhead here to study the transferability of model predictivity. * denotes two easier settings where all methods obtain near-zero average gaps over $T_{\text{limit}} = 60$s, so we instead run all methods for a shorter $T_{\text{limit}} = 10$s.

| Target | Source | Unguided | Linear | Per-Subset | Multi-Subset |
|---|---|---|---|---|---|
| empty, 350, U50 | empty, 350, U50
random, 250, A25 | 42 ± 1 | 39 ± 1
40 ± 1 | 43 ± 1
41 ± 1 | **30 ± 1**
**37 ± 1** |
| empty, 350, A25 | empty, 350, U50
random, 250, A25 | 42 ± 1 | 38 ± 1
39 ± 1 | 39 ± 1
40 ± 1 | **28 ± 0.8**
**37 ± 1** |
| empty, 300, A25* | empty, 350, U50
random, 250, A25 | 28 ± 1 | 36 ± 2
29 ± 1 | 36 ± 2
27 ± 1 | **24 ± 1**
**22 ± 0.9** |
| empty, 250, A25* | empty, 350, U50
random, 250, A25 | 10 ± 0.5 | 14 ± 1
11 ± 0.7 | 14 ± 1
9.8 ± 0.6 | **9.4 ± 0.7**
**8.0 ± 0.4** |

difficult as Unguided does not require a GPU, we believe that research into deep learning-based acceleration in problems like MAPF is worthwhile due to the rapid advances in GPU technology (our own NVIDIA V100 is two generations old). To our knowledge, despite copious research in MAPF, there exists no other work leveraging deep learning for enhancing state-of-the-art MAPF algorithms at our scale. In addition, our work demonstrates the potential of deep-learning-guided LNS beyond graphical problems and encompasses problems with complex spatiotemporal constraints, where it may serve as a blueprint for future learning-based iterative methods. Our convolution-attention blocks may be more broadly applicable for representing pathwise interactions beyond LNS settings. Some immediate extensions of our work include guiding stochastic solvers like prioritized planning. Furthermore, the Multi-Subset architecture may allow direct construction of subsets for LNS rather than evaluating heuristically-constructed subsets. Finally, future work may investigate the effectiveness of unguided and guided LNS techniques in real-world robotic warehouses.

## 8 ACKNOWLEDGMENT

The authors acknowledge the MIT SuperCloud and Lincoln Laboratory Supercomputing Center for providing HPC resources that have contributed to the research results reported within this paper. This work was supported by the MIT Amazon Science Hub as well as a gift from Amazon.

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

# A    APPENDIX

## CONTENTS

## A.1    SETUP

We discuss the full experimental setup here for all experiments performed in our paper, unless already discussed in Section 5.

**Warehouses**    The MAPF benchmark suite Stern et al. (2019) is a set of empty, random, maze-like, game-like, and city-like floor maps. Here we describe the floor maps presented in Figure 4:

1. empty-32-32: a size $32 \times 32$ map with no static obstacles.
2. random-32-32-10: a size $32 \times 32$ map with randomly sampled static obstacles covering 10% of the vertices.
3. warehouse-10-20-10-2-1: a size $161 \times 63$ procedurally generated warehouse with static obstacles covering 44% of the vertices.
4. ost003d: a size $194 \times 194$ map taken from the game Dragon Age Origin, with obstacles covering 65% of the vertices.
5. den520d: a size $256 \times 257$ map taken from the game Dragon Age Origin, with obstacles covering 57% of the vertices.

The starts and goals are each selected uniformly at random without replacement from the set of non-obstacle vertices. All floor maps are undirected graphs

**Architecture Parameters**    For the Per-Subset architecture, we use two 3D convolutional blocks (convolution, batch normalization, ReLU, max pooling) with 32 and 64 channels, respectively, followed by two 2D convolutional blocks with 128 channels each. For the Multi-Subset architecture, we use eight convolution-attention blocks with $(C =)$ 16, 16, 32, 32, 64, 64, 128, and 128 channels, respectively, shared across all subsets, followed by three Transformer multi-head attention blocks with 128 features for each subset. A time cutoff of $T = 48$ is used for both architectures in empty (32x32) and random (32x32), $T = 96$ for warehouse (161x63), $T = 192$ for ost003d (194x194), and $T = 216$ for den520d (256x257). Pre-applied temporal pooling with factors 3, 6, and 6 are used for Per-Subset for warehouse, ost003d, and den520d, respectively; pre-applied temporal poooling with factors 2, 4, 4 are used for Multi-Subset, respectively. Pre-applied spatial pooling with factors 2, 4, and 4 are used for Per-Subset, respectively. Pre-applied spatial pooling with factors 4 are used for Multi-Subset for ost003d and den520d.

**Loss Function for Per-Subset Architecture**    Unlike Multi-Subset and Linear, the Per-Subset architecture cannot efficiently encode large multiples of $J$ subproblems simultaneously. Therefore, rather than a pairwise classification loss, we utilize a clipped mean squared error loss. Denoting $f_\theta := f_\theta(P(S^i, \alpha_j))$ to be the neural network's output score for subproblem $j$ and $\delta_{\text{Solver}} := \delta_{\text{Solver}}(S^i, \alpha_j)$ to be the ground-truth improvement, this loss function is

$$\ell(f_\theta \mid S^i, \alpha_j) = (f_\theta - \min(\text{stop\_grad}(f_\theta), 0) - \max(\delta_{\text{Solver}}, 0))^2 \qquad (2)$$

with gradient

$$-\frac{\mathrm{d}\ell(f_\theta \mid S^i, \alpha_j)}{\mathrm{d}f_\theta} = -2\left(f_\theta - \min(\mathrm{stop\_grad}(f_\theta), 0) - \max(\delta_{\mathrm{Solver}}, 0)\right)$$

$$= 2\begin{cases} \delta_{\mathrm{Solver}} - f_\theta, & \text{if } f_\theta > 0 \text{ and } \delta_{\mathrm{Solver}} > 0, \\ -f_\theta, & \text{if } f_\theta > 0 \text{ and } \delta_{\mathrm{Solver}} \le 0, \\ \delta_{\mathrm{Solver}}, & \text{if } f_\theta \le 0 \text{ and } \delta_{\mathrm{Solver}} > 0, \\ 0, & \text{if } f_\theta \le 0 \text{ and } \delta_{\mathrm{Solver}} \le 0, \end{cases}$$

which only encourages $f_\theta$ to be non-positive rather than forcing $f_\theta$ to approach $\delta_{\mathrm{Solver}}$ when $\delta_{\mathrm{Solver}} < 0$; this allows extra network capacity to be dedicated to fitting positive improvements well, rather than unnecessarily fitting negative improvements.

Additional clipping similar to that in Equation 2 is also performed for the loss in Section 4.4.

**Training**   We train all neural models with the Adam optimizer Kingma and Ba (2014), decaying learning rate (0.01 for Per-Subset and 0.0001 for Multi-Subset) with cosine annealing across 100000 training steps. Per-Subset architecture sees a minibatch of $512$ subsets per gradient step, and Multi-Subset architecture sees a minibatch of $16J = 1600$ subsets per step. Training takes roughly 24 hours on a single NVIDIA V100 GPU. For the Linear baseline, we found the Scikit-learn Pedregosa et al. (2011) implementation of linear support vector machine (SVM) Fan et al. (2008) with full-batch gradient descent to outperform our minibatched gradient descent-based training loop; we sub-sample the subsets as required to stay within the batch size limits of linear SVM (note that the total batch size scales with $J^2$). Training for the Linear baseline takes around 10 minutes to 1 hour on a single 48-CPU Intel Xeon Platinum 8260 processor. Hyperparameters of all models are manually tuned on one experimental setting, then replicated across all other experimental settings.

**Validation**   During training, we periodically validate our learned model's predictivity on a validation set with held-out seeds. Predictivity is measured by a collection of proxy metrics, such as correlation between predicted scores and ground-truth improvements, rather than directly running guided LNS with the current model checkpoint, which would take more time. The checkpoint with the best validation performance (typically the one at the end of training) is selected, and further validation may be performed for additional hyperparameter selection by guiding LNS with the selected checkpoint.

**Test**   The final selected model from validation is tested on 200 additional held-out random seeds by guiding LNS. All reported results are performances on the test set.

**Guiding LNS**   For each given seed, all methods (unguided or guided) share the same initialization solution. The planning time limit of $T_{\mathrm{limit}} = 60$s is used by previous LNS-based MAPF work (Li et al., 2021a; Huang et al., 2022) and is motivated by the short-horizon nature of MAPF and path planning in general. We observe that learned models may often repeatedly select similar subsets across consecutive LNS iterations, resulting in low or no improvement. As such, we design a stratified sampling procedure where we filter away any subset more similar to previously selected subsets than a randomly sampled threshold, and permit the learned model to select among the rest of the subsets. We observe that this procedure is effective in alleviating the similarity issue.

**Fitting a Proxy for Solver Runtime**   As the solver's runtime is noisy to measure and heavily depends on the current computation load of the machine, we instead collect datasets of number of low-level search iterations by the solver as features and ideal solver runtimes as labels. For every setting (floor map, $|A|$, construction heuristic, $k$), features and ideal solver runtimes are obtained by running unguided LNS with no other processes on the machine. We obtain fitted linear models with 0.95-0.99 coefficient of determination across five folds of cross-validation.

**Aggregating Metrics Across Seeds**   We always report the 95% confidence interval of the mean across the 200 test seeds, though any seed without feasible initialization is excluded. *e.g.* to compute the 95% confidence interval of the average gap at a given runtime, we perform bootstrap sampling to obtain 1000 resampled means of 200 gaps, then take the 2.5th percentile and 97.5th percentile

of those means as the 95% confidence interval. To compute the mean and confidence interval of speedups corresponding to different seeds at a given gap, we take the geometric mean rather than the arithmetic mean, as each speedup is a ratio. When plotting speedup vs gap, some seeds may not reach a given gap within 10s of runtime; therefore, we only consider the range of gaps reached by all seeds when plotting, rather than $[0, 100]$.

## A.2 PARAMETER SWEEP FOR SOLVER, SUBSET CONSTRUCTION HEURISTICS, AND SUBSET SIZE $k$

We perform sweeps to identify the strongest settings for Unguided LNS for the different warehouse maps and use these settings for guided LNS. As seen in Figure 6, PBS is fairly dominant for most empty (32x32) settings with $|A| < 400$. On the other hand, PP with $k = 5$ becomes dominant for $|A| \geq 400$. In our work, we experiment with settings where PBS is dominant, since a deterministic solver could be easier to predict for learning-based methods. As seen in Figure 7, PBS is the dominant solver across a wide range of problem settings anyways.

Following Li et al. (2021a), we additionally sweep over $k$ with Adaptive LNS (ALNS). We do not see benefits of ALNS over the best fixed subset construction in random, empty, warehouse, and den520d. Only ost003d benefits slightly from ALNS. This is fairly consistent with the findings of Li et al. (2021a) Table 2. Thus, we choose to study the more controlled setting of best fixed subset construction rather than adaptive subset construction.

Unlike Huang et al. (2022), we do not select between subsets of different sizes $k$. Intuitively, subproblems with different sizes could require significantly different amount of time to solve by the solver. Neither our work nor MAPF-ML-LNS devises a method to account for the runtime of solving subproblems; purely selecting the subset offering the most improvement may be highly suboptimal if the subproblem takes relatively longer to solve than other candidates. Thus, we choose to use a fixed subset size $k$, assuming that subsets of the same size likely take similar amount of time to solve. This assumption is confirmed by our iteration-based results in Appendix A.8. Our method is likely compatible with future LNS methods designed to account for both predicted improvement and predicted runtime of subsets.

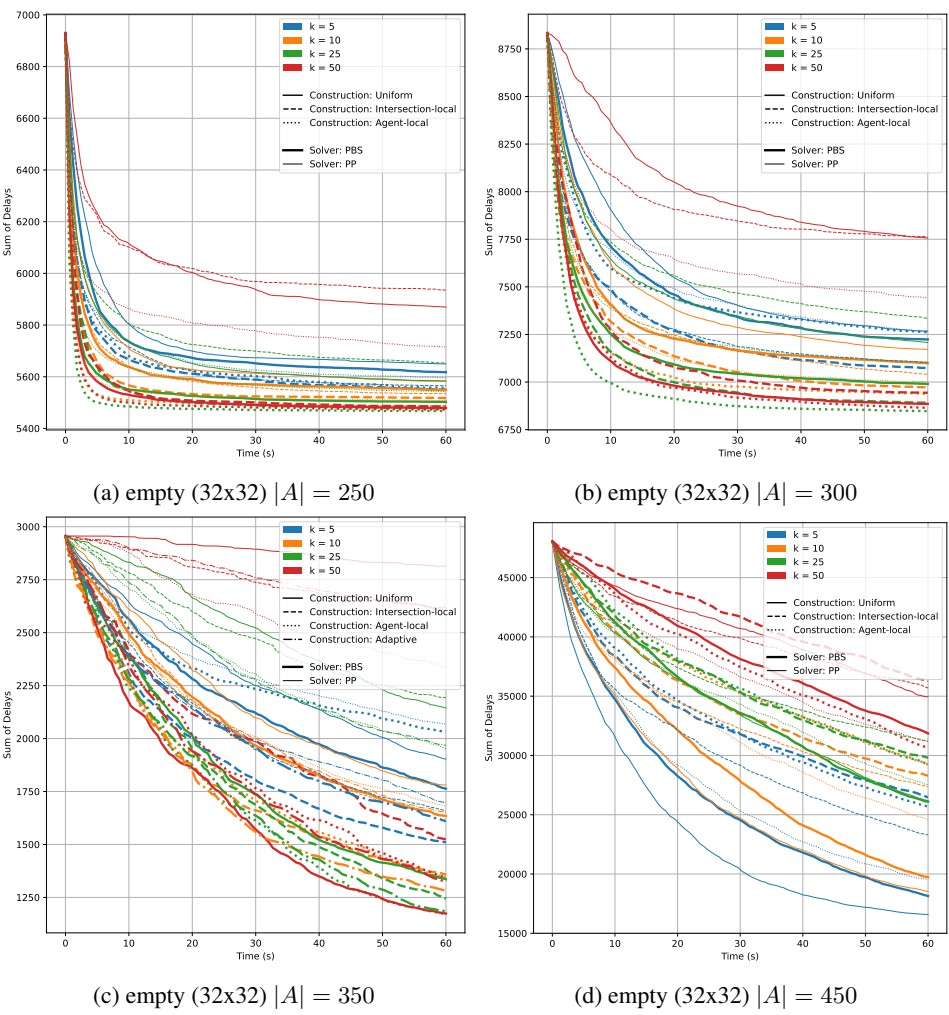

(a) empty (32x32) $|A| = 250$

(b) empty (32x32) $|A| = 300$

(c) empty (32x32) $|A| = 350$

(d) empty (32x32) $|A| = 450$

Figure 6: **Effect of solver, subset construction heuristics, and agent subset size $k$ in unguided LNS for empty (32x32).** Colors distinguish subset sizes $k$, line styles distinguish construction heuristics, and line widths distinguish solvers. The Adaptive construction heuristics adaptively chooses between Uniform, Intersection-local, and Agent-local heuristics following Li et al. (2021a).

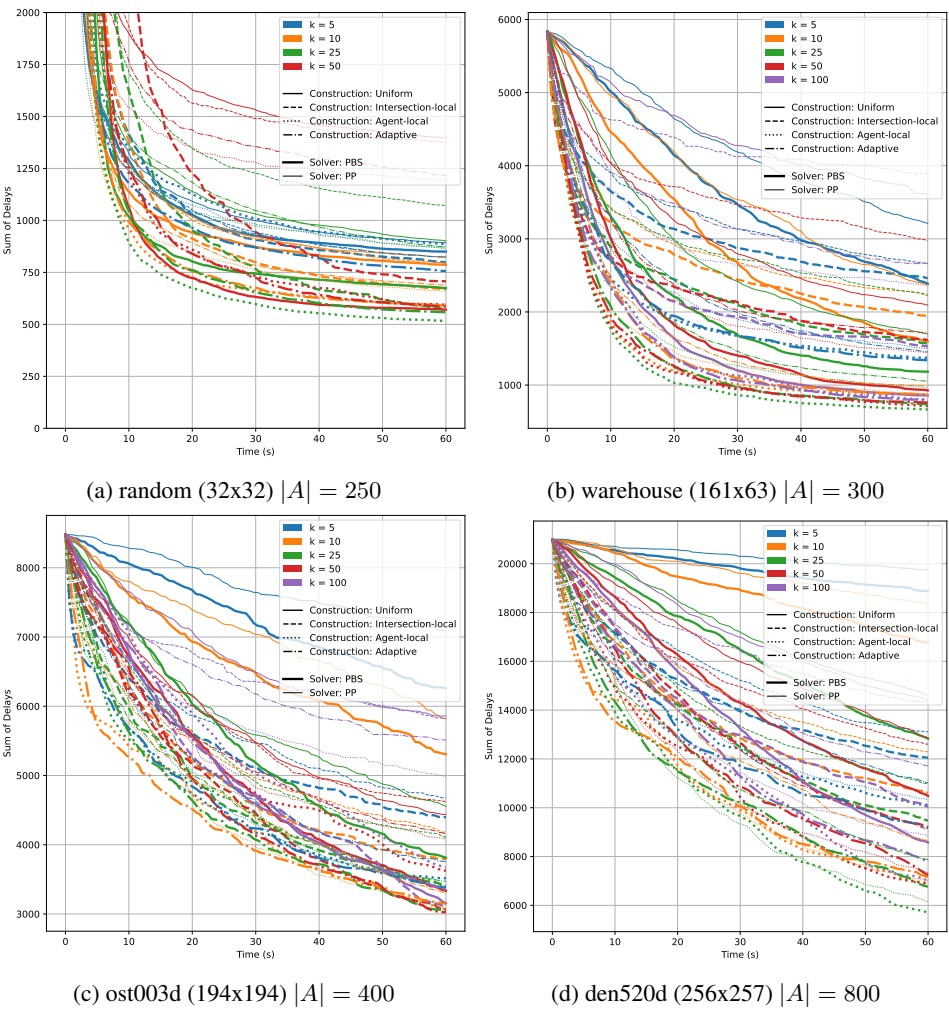

(a) random (32x32) $|A| = 250$

(b) warehouse (161x63) $|A| = 300$

(c) ost003d (194x194) $|A| = 400$

(d) den520d (256x257) $|A| = 800$

Figure 7: **Effect of solver, subset construction heuristics, and agent subset size $k$ in unguided LNS.** Colors distinguish subset sizes $k$, line styles distinguish construction heuristics, and line widths distinguish solvers. The Adaptive construction heuristics adaptively chooses between Uniform, Intersection-local, and Agent-local heuristics following Li et al. (2021a).

## A.3 RUNTIME VS SOLUTION QUALITY FOR ALL SETTINGS

Here, we report the full results corresponding to Figure 5 in the main text.

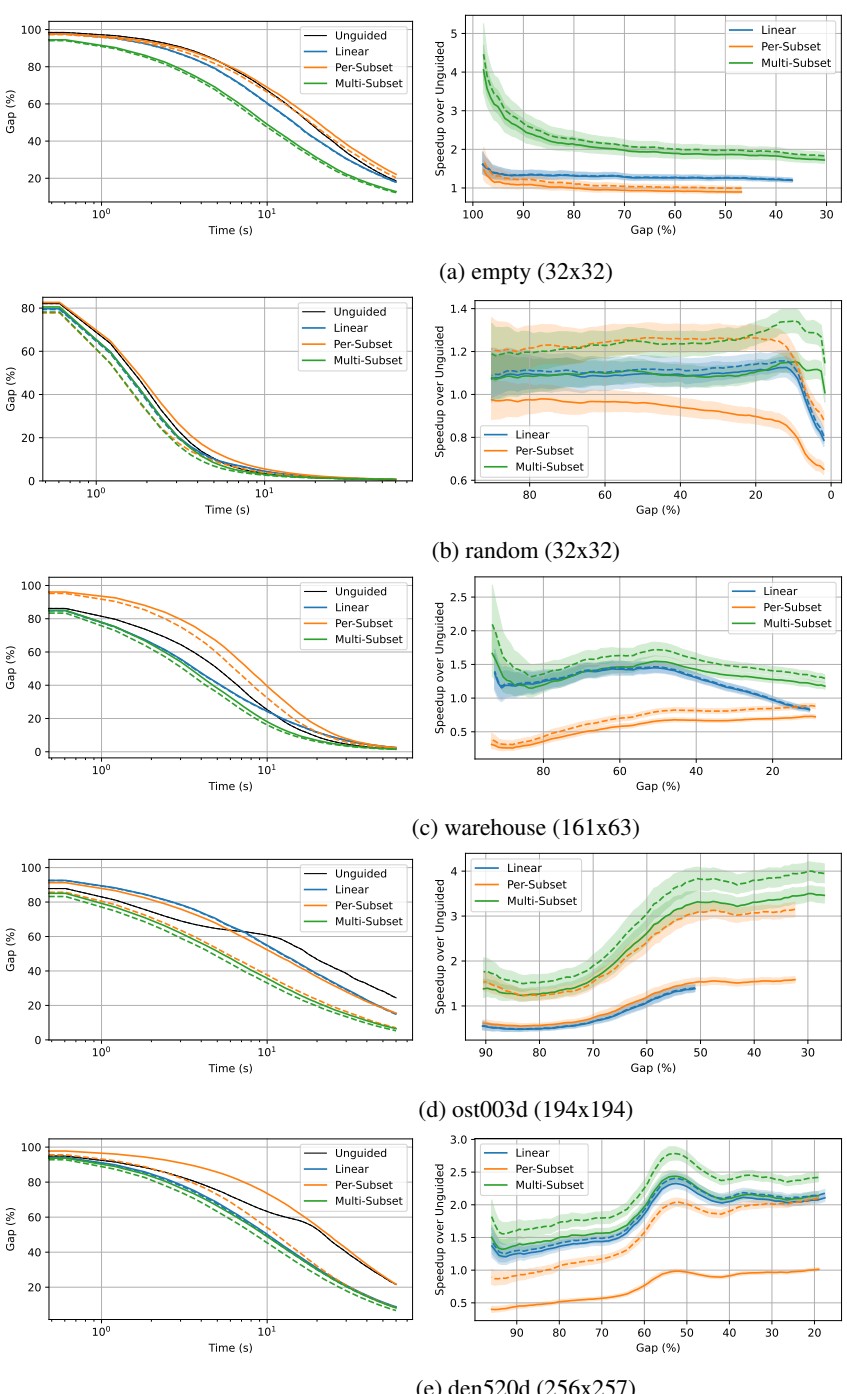

Figure 8: **Runtime vs solution quality for all considered floor maps** including (solid) and excluding (dashed) model overhead in the runtime. (left) Lower gap is better. (right) Higher speedup over Unguided is better.

### A.4 Handcrafted Features for Linear Baseline

The features from our Linear baseline are hand-designed by Huang et al. (2022) and reiterated here. Given current solution $S^i$, to featurize subproblem $P(S^i, \alpha)$, features are calculated for every agent $a \in A$ then combined into aggregate features for the subset $\alpha$ and the non-subset $A \setminus \alpha$, which are then concatenated.

Table 3 lists the 16 features (Huang et al. (2022) states 17, but only lists 16) for each agent $a$, consisting of 6 static features which only depend on the problem $P = (G, s_A, g_A)$ and 10 dynamic features which depend on the current solution $S^i$. The *heat* of a vertex $v \in V$ is defined to be the total number of times the vertex is occupied by an agent. The *degree* of a vertex $v \in V$ is defined to be the total number of edges to or from $v$, excluding the self-edge $(v, v)$. The per-agent features are then aggregated with minimum, maximum, sum, and average across $\alpha$ and $A \setminus \alpha$, resulting in $4 * 2 * 16 = 128$ total features for the subset $\alpha$.

We vectorize the computation of these features to allow faster feature-computation on GPU, as the CPU-only feature computation tends to be several times slower.

Table 3: **Linear features from Huang et al. (2022) for each agent** $a$**.** These features are then aggregated by subset $\alpha$ and non-subset $A \setminus \alpha$.

| *Description* | *Count* |
|---|---|
| **Static features** | 6 |
| Distance between start and goal: $d(s_a, g_a)$ | 1 |
| Row and column numbers of $s_a$ and $g_a$ in graph $G$ | 4 |
| Degree of the goal $g_a$ | 1 |
| **Dynamic features** | 10 |
| Delay of $a$: $c(S_a)$ | 1 |
| Ratio between delay and shortest distance: $\frac{c(S_a)}{d(s_a, g_a)}$ | 1 |
| Minimum, maximum, sum, and average of heat values along path $p_a$ | 4 |
| Number of timesteps that path $p_a$ passes a vertex with degree $1 \leq j \leq 4$ before reaching the goal $g_a$ | 4 |

### A.5 Software Framework and Code

A key challenge of studying learning-based approaches for MAPF, especially LNS-based, is the low-level nature of MAPF, where the solvers must be implemented in a low-level language like C++. Performing (especially learning-based) experiments within such a context is tedious and error-prone for the researcher. We design a software framework which abstracts away low-level MAPF operations into a C++ library, exposing a high-level interfaces which can be controlled through Python via a convenient pybind11 interface while maintaining execution speed. We hope that such a framework could prove useful for MAPF researchers in general, greatly reducing the need to interact with low-level C++.

Table 4: **Architectural ablations** in random (32x32) $|A| = 250$. Gaps and Win / Loss do not include model overhead. Lower gap is better; higher Win is better.

| Metric | Unguided | Linear | Per-Subset | Multi-Subset | | |
| --- | --- | --- | --- | --- | --- | --- |
| | | | | (Full) | (Convolution Only) | (Attention Only) |
| Average Gap (%) | 5.3 ± 0.2 | 5.4 ± 0.3 | 5.0 ± 0.2 | **4.6 ± 0.2** | 6.6 ± 0.3 | 5.2 ± 0.2 |
| Win / Loss | 0 / 0 | 106 / 92 | 128 / 70 | **151 / 47** | 27 / 171 | 112 / 86 |

### A.6 ARCHITECTURAL ABLATIONS

To assess the importance of our proposed intra-path attention mechanism as a part of the convolution-attention block in the Multi-Subset network, we conduct an ablation study to understand the individual contributions of intra-path attention and 3D convolution in our architecture. For Multi-Subset (3D Convolution Only), we remove the intra-path attention mechanism and double the number of 3D convolutions to keep model capacity similar. For Multi-Subset (Intra-path Attention Only), we remove the 3D convolution and double the number of intra-path attention layers; however this requires us to use $(C =)$ 128 channels at every layer (increasing the model capacity) as the intra-path attention itself cannot change the number of channels from layer to layer. Due to the changes in architecture, we compare performances without model overhead to focus on the predictivity of the different architectures.

In Figure 9, we see that our intra-path attention is critical to Multi-Subset performance, while 3D convolution also contributes moderately. Corresponding metrics are reported in Table 4.

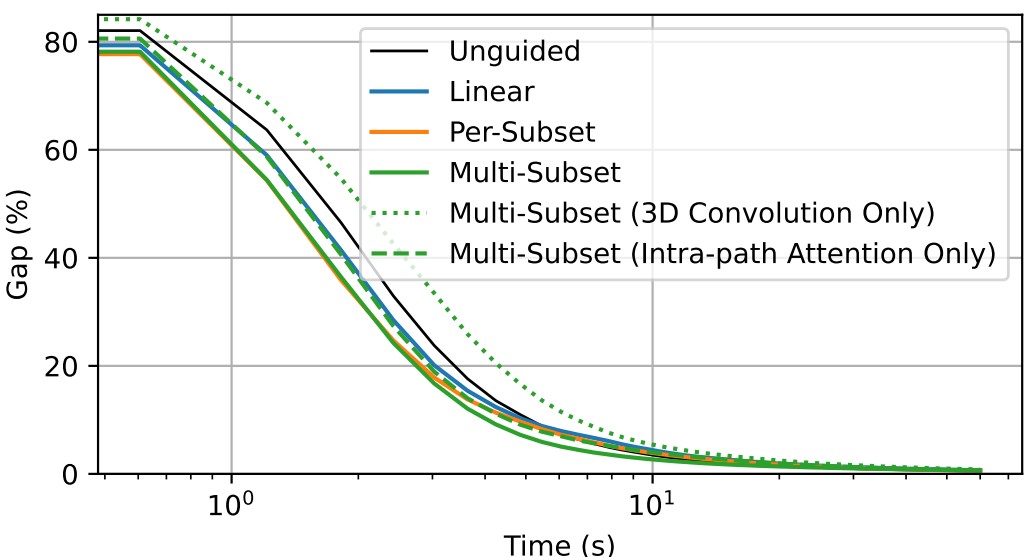

Figure 9: **Runtime vs gap for architectural ablation in random (32x32)** $|A| = 250$, excluding model overhead in the runtime. Lower gap is better.

Table 5: **Absolute "sum of delays" comparisons with MAPF-LNS (Li et al., 2021a) and MAPF-ML-LNS (Huang et al., 2022).** LNS, LNS (ML-LNS), and Unguided denotes the best unguided LNS as respectively reported by Li et al. (2021a), Huang et al. (2022), and our work. ML-LNS and Linear denotes the best linear-guided LNS as respectively reported by Huang et al. (2022) and our work.

| Setting | LNS | LNS (ML-LNS) | Unguided | ML-LNS | Linear | Multi-Subset |
|---|---|---|---|---|---|---|
| empty $|A| = 350$ | **743** | N/A | 1302 | N/A | 1263 | 1043 |
| random $|A| = 250$ | 3388 | 806 | 516 | 843 | 532 | **515** |
| warehouse $|A| = 300$ | 1400 | 4719 | 670 | 3547 | 686 | **658** |
| ost003d $|A| = 400$ | 2427 | 6907 | 2736 | 6584 | 2119 | **1760** |
| den520d $|A| = 800$ | 7408 | ≫12558 | 4637 | ≫11535 | 2397 | **2229** |

A.7 DIRECT COMPARISONS WITH MAPF-LNS AND MAPF-ML-LNS

While our computation resources may differ from previous related works (Li et al., 2021a; Huang et al., 2022), we attempt to provide absolute comparisons of sums of delays attained by all methods in Table 5. The sum of delays is a measure of final solution quality: the final cost attained minus the shortest individual cost for each agent. The sums of delays of all methods here are attained with 60s of solution time including the model overhead. Like previous work, here we use the 25 seeded scenarios provided by the MAPF benchmark suite Stern et al. (2019). For our Unguided, Linear, and Multi-Subset, we use the same hyperparameters as in Table 1, which represent the best fixed subset construction heuristic and subset size $k$. We take the best MAPF-LNS (Li et al., 2021a) sums of delays as the minimum of their Table 4, Table 2, and Table 1. We take MAPF-ML-LNS (Huang et al., 2022) sums of delays from their Table 3.

We make several observations. Our numerical sums of delays here are fairly consistent with our hyperparameter sweeps in Appendix A.2, which uses a different set of seeds. Comparing the baselines, MAPF-LNS's own reported sums of delays (denoted as LNS in Table 5) are usually significantly lower than MAPF-LNS sums of delays reported by the MAPF-ML-LNS paper (denoted as LNS (ML-LNS) in Table 5), as seen in warehouse, ost003d, and den520d. Our Unguided sums of delays are often significantly lower (random, warehouse, den520d) than those of MAPF-LNS, illustrating the advantage of using PBS rather than PP as the subproblem solver, which has not been studied by previous works; indeed, the runs with PP as the subproblem solver in our hyperparameter sweeps in Figure 7 more closely resemble the best reported MAPF-LNS performance. Our Linear baseline significantly outperforms MAPF-ML-LNS in all settings. Multi-Subset improves the sums of delays over Unguided, especially in empty, ost003d, and den520d.

We point out two cases where our sums of delays are worse than MAPF-LNS. For empty $|A| = 350$, MAPF-LNS moderately outperforms Unguided and Multi-Subset. We hypothesize that this is because Li et al. (2021a) uses EECBS (Li et al., 2021c) as the initialization for this particular setting while we use PP, which is easier to implement in our codebase than EECBS (which contains many hyperparameters itself). For ost003d $|A| = 400$, MAPF-LNS slightly outperforms Unguided (but not Multi-Subset). We observe in Figure 7 that adaptive LNS slightly improves the performance of Unguided in this setting. Nevertheless, the difference between MAPF-LNS and Unguided is dwarfed by the difference between Unguided and Multi-Subset.

Overall, we find that our methods compare reasonably with prior work. We additionally illustrate corresponding time vs delay in Figure 10.

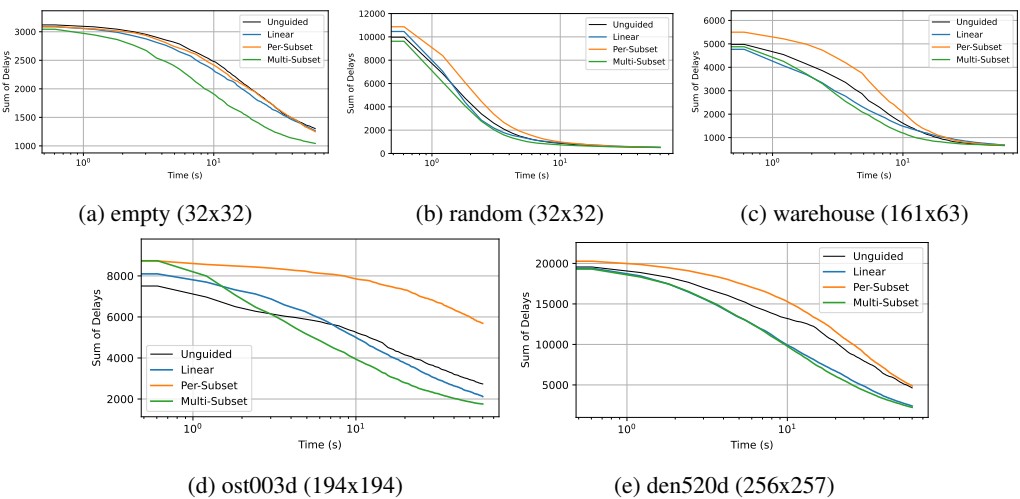

Figure 10: **Runtime vs sum of delays for all considered floor maps** including model overhead in the runtime. Lower sum of delays is better.

## A.8 PERFORMANCE BY ITERATION

For the same runs studied in Appendix A.7, we demonstrate the sum of delays with respect to number of LNS iterations in Figure 11. We see similar behavior as in Figure 10: that the Multi-Subset architecture significantly improves the decision quality and reduces the number of iterations needed to attain a given cost. The similarities in the plots also confirm our assumption that subsets of the same construction heuristic and size $k$ take roughly similar time to solve.

Note that because we run each method for 60s, the number of iterations may differ between problem instances. As we must compute the mean and confidence interval of the performance across all seeds for each method, we truncate all runs at the minimum number iterations across all seeds for each method when calculating the mean and confidence interval. This choice does not have a significantly visible effect due to the log-scale of LNS iteration in Figure 11.

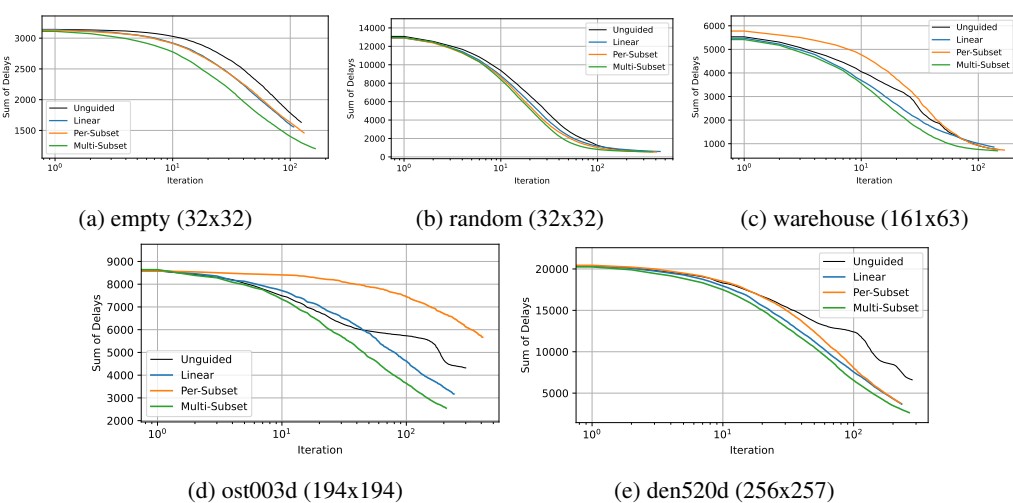

Figure 11: **LNS iteration vs sum of delays for all considered floor maps**. Lower sum of delays is better.

