# OpenReview forum: "Neural Neighborhood Search for Multi-agent Path Finding"
_ICLR.cc/2024/Conference — ICLR 2024 poster_

### Official Review · Reviewer_pDBe · 2023-10-22

**Soundness:** 3 good
**Presentation:** 4 excellent
**Contribution:** 3 good
**Rating:** 6
**Confidence:** 3

**Summary:**

This paper presents a novel approach for leveraging machine learning in multi-agent path planning (MAPF) based on large neighbor search (LNS). Given an initial solution for a MAPF problem instance, LNS selects a subset of agents and optimizes their solution paths, treating the paths of other agents as spatiotemporal obstacles.

Traditionally, subset selection has relied on heuristic rules or linear models with hand-crafted features. However, the method proposed here integrates a deep neural network into the agent subset selection process. This network predicts the performance gain achievable by selecting a particular subset, using a tensor that encapsulates the agents' current paths, potential shortest paths, and the layout of other obstacles. To manage the computational expense of applying this procedure to various subsets, the proposed method initially extracts features from all agents' paths and predicts the final gain from the feature map slices corresponding to the subset, facilitated by the deep network.

The method's effectiveness has been validated on MAPF problems involving hundreds of agents.

**Strengths:**

**Clarity**: The paper is very well-written overall, and it accurately positions the proposed method against various related studies.

**Novelty**: The structure of the network used to estimate performance gain for agent subset selection and the method of constructing input tensors are new and intriguing.

**Quality**: Overall, the work is of high quality. The proposed method appears solid and technically sound. The authors have implemented numerous techniques to reduce computational costs, a critical factor in MAPF where execution time is crucial.

**Significance**: The method's significance has been evaluated across multiple MAPF problems, and it has been tested on scenarios involving hundreds of agents. It is commendable that the proposed method can handle hundreds of agents, though I still have some concerns about experimental results, as shown below.

**Weaknesses:**

Despite the paper's strengths, it could benefit from a more persuasive quantitative evaluation, which would likely enhance its impact.

In Table 1, the difference between the Linear baseline and the proposed method appears relatively marginal. While the proposed method outperforms the Linear baseline in terms of average gap and final gap scores, the Linear baseline requires less runtime overhead. Given this trade-off between computation time and solution quality, how can the benefits of the proposed method be demonstrated?

Additionally, it is unclear what computational resources the proposed method and baseline methods require. The paper suggests that both the proposed method and Linear baseline were tested on a GPU, but it doesn't specify the details of GPU specs required. In practical scenarios, not all environments have access to high-end GPUs, and being able to execute path planning on affordable entry-level GPUs or standard CPUs could be a significant advantage.

**Questions:**

While the paper is well-written and the proposed method is clearly explained, there are some uncertainties regarding the evaluation experiments.

Specifically, how can we demonstrate the effectiveness of the proposed method over the Linear baseline? While the solution quality obtained by the proposed method is better, the Linear baseline can run faster instead. Moreover, I wonder if the proposed method may require much more GPU resources than other methods and could be much slower when performed on CPUs.

For example, is it possible to reduce the model capacity for the proposed method so that it can run as approximately fast as the Linear baseline, and compare the solution quality under such conditions?

---

> ### Author Response · Authors · 2023-11-15
>
> We thank the reviewer for the feedback and suggestions and encourage the reviewer to view our general response to all reviewers, which contain significant improvements to our paper. Here are our specific responses to each concern:
>
> **Weaknesses**
>
> 1. We would like to clarify that the Average Gap (%), Win / Loss, and Final Gap (%) in Table 1 **already** include the stated model overhead in the 60s runtime (they are exactly the solid curves in Figure 5). Thus, our Multi-subset architecture outperforms Linear in all settings (especially in empty, random, warehouse, and ost003d), even with higher overhead. We believe that this fairly demonstrates the benefits of our method.
> 2. Linear, Per-subset, and Multi-subset all use a single NVIDIA V100 GPU (released in 2017, Volta architecture) rather than more recent top-end GPUs. Our understanding is that recent affordable commodity GPUs such as NVIDIA RTX 3060 (released in 2021, Ampere architecture) perform similarly to the NVIDIA V100.
>
> **Questions**
>
> As mentioned above, we do include inference overhead for both our method and the Linear baseline. We agree that our method would be slower than Linear if executed on a CPU. We agree that standard model compression techniques may reduce our model overhead significantly to be more runnable on a CPU, but this is orthogonal to our proposed method and can be further investigated by practitioners. For example, we could imagine that our Multi-subset model could serve as a teacher model for a Linear student model during model distillation.
>
> We thank the reviewer for their time. We hope that the reviewer could update our score if we have further strengthened our paper through our additional rebuttal efforts.

---

> > ### Comment · Reviewer_pDBe · 2023-11-15
> > **Thank you**
> >
> > Thank you for the response!
> >
> > Ok I understood that from Fig 5 the proposed method is more efficient than Linear in terms of how quickly the gap is improved.
> > My points have been resolved, but I just wanted to make clear what the "Overhead (s)" in Table 1 actually means, as it seems to be less explained in the main text. For example, for the "empty" result, the overhead is +0.002 for Linear and +0.013 for Multi-Subset, which I understood that the Linear baseline has lower overhead.

---

> > > ### Comment · Reviewer_pDBe · 2023-11-21
> > > **Rating**
> > >
> > > Based on the clarification above I have updated my rating.

---

> > > > ### Author Response · Authors · 2023-11-23
> > > >
> > > > We are happy to see that the reviewer's concern has been addressed. We again thank the reviewer for taking the time to review our work and look over our responses!

---

### Official Review · Reviewer_TyGL · 2023-10-29

**Soundness:** 2 fair
**Presentation:** 3 good
**Contribution:** 2 fair
**Rating:** 3
**Confidence:** 4

**Summary:**

The paper addresses Large Neighborhood Search (LNS) for Multi-Agent Pathfinding (MAPF) using deep learning. Starting with a feasible (but suboptimal) solution, LNS can solve MAPF in an anytime manner by iteratively destroying and repairing parts, i.e., the neighborhood, of the incumbent solution to improve the solution quality over time. The neighborhoods are determined by using some heuristics from the literature. Given a set of neighborhood candidates, the paper proposes to use deep learning to select a suitable neighborhood via score prediction. The deep learning approach exploits the spatio-temporal structure of the MAPF problem by encoding the paths and obstacles in separate tensors, which are processed by a series of 3D and 2D convolutional layers as well as an attention mechanism. The approach performs well compared to state-of-the-art approaches in selected problems and displays some generalization capabilities for maps of the same size.

**Strengths:**

The paper addresses an interesting application area that is well-known in the AI community.

The paper is well-written and easy to understand.

**Weaknesses:**

**Novelty**

The main contribution of the paper is the application of standard deep learning techniques to 2D grid world MAPF. Since tensor encoding of grid worlds is common practice [1,2,3], I do not consider the approach as particularly novel. The architecture is common practice as well (standard convolutional layers and attention), where the application to a new domain seems to be the main contribution to me.

[1] D. Silver et al., "Mastering the Game of Go with Deep Neural Networks and Tree Search", Nature 2016

[2] J. Leibo et al., "Multi-Agent Reinforcement Learning in Sequential Social Dilemmas", AAMAS 2017

[3] M. Dennis et al., "Emergent Complexity and Zero-shot Transfer via Unsupervised Environment Design", NeurIPS 2020

**Soundness**

As noted in the paper, MAPF-LNS relies on fast operations (like prioritized planning/PP for repairing and linear models for neighborhood scoring) to ensure its success as an anytime algorithm. However, the paper proposes several modifications to the standard/default setting of the MAPF-LNS or MAPF-ML-LNS paper, which actually increase runtime:
- A relatively large model for score prediction compared to simple linear models
- Generation of several neighborhoods at each iteration (which requires the invocation of the destroy heuristics several times)
- Priority-based planning (PBS) for repairing, which is slower than the default PP due to backtracking in the tree search

MAPF-LNS and MAPF-ML-LNS rely on a large number of iterations to achieve a good solution quality. Therefore, I am not sure if the addition of several more expensive operations actually pays off since it should significantly limit the number of possible iterations.

The generalization depends on the map size. A model trained on $32 \times 32$ maps cannot be straightforwardly used on, e.g., $24 \times 64$ maps and is therefore limited.

**Significance**

The paper evaluates with different hyperparameters than suggested in the original literature [4,5]. Thus, I am
1. uncertain about the fairness of the evaluation and
2. skeptical about the effectiveness, since some changes increase runtime that would limit the number of iterations for sufficient search (see above).

I am also not sure if the modifications used in the deep learning variant are also applied to the linear version. If so, the comparison might be unfair since MAPF-ML-LNS uses some mechanisms, e.g., random neighborhood sizes, that are seemingly important for its success.

The experiments only report relative numbers, which makes it difficult to relate to the performance reported in prior work, e.g., do the baselines still perform similarly? In that case, fairness could be confirmed, at least.

The evaluation only considers maps with a fixed number of agents; therefore, I have no intuition on how the average/final gap would scale, e.g., with an increasing number of agents.

[4] J. Li et al., "Anytime Multi-Agent Path Finding via Large Neighborhood Search", IJCAI 2021

[5] T. Huang et al., "Anytime Multi-Agent Path Finding via Machine Learning-Guided Large Neighborhood Search", AAAI 2022

**Evaluation**

To decide whether I raise my score or not, I first need to check the following:
- Since MAPF algorithms are generally very implementation-dependent (as stated in the appendix), I need to confirm the validity of the proposed mechanisms by viewing and running the code myself (with a provided trained model).
- I need to see plots or tables with the absolute performance, i.e., the sum of delays, for different numbers of agents per map. The evaluation can be easily done by running the experiments with the neural LNS on the exact same setting as [5] and comparing it with the performance of MAPF-LNS and MAPF-ML-LNS reported in that paper.
- I need to see plots or tables with the number of iterations and success rate per iteration for different time budgets. If the approach was valid, we should see a lower iteration count than the state-of-the-art but a higher success rate.

**Questions:**

- *“we perform parameter sweeps detailed in Appendix A.2 to first identify the strongest possible configuration for the Unguided baseline”* - The original MAPF-LNS paper already reported extensive hyperparameter experiments to determine good hyperparameters. Why was it necessary to tune them again?
- I wonder why deep learning was only used for neighborhood selection via scoring, while the neighborhood generation is still based on the handcrafted destroy heuristics. Wouldn't it make sense to address the generation via deep learning as well to make the approach more end-to-end [6]?

[6] Y. Wu et al., "Learning Large Neighborhood Search Policy for Integer Programming", NeurIPS 2021

---

> ### Author Response · Authors · 2023-11-15
> **Response Part 1/3**
>
> We are very grateful for the reviewer’s suggestions to improve the rigor and soundness of our experiments by providing additional perspectives of performance. We encourage the reviewer to first examine our general response to all reviewers, where we summarize major rebuttal updates. Here we focus on each comment in more detail:
>
> **Novelty**: We discuss the novelty of our proposed intra-path attention in the general response and its aptness for multi-agent path planning. We encourage the reviewer to view updated architectural figure (Figure 3) as well as additional ablation (Appendix A.6).
>
> **Soundness**:
> Regarding the reviewer’s comments about sources of increase in runtime:
>
> 1. Model overhead: the neural network overhead is indeed larger than Linear. However, what’s important is the *relative overhead*; that is, the model overhead is insignificant when it is small compared to the repair heuristic. In Table 1, we observe this phenomenon in many cases for the Multi-Subset network, especially in large settings like den520d (256x257). In real-world deployment settings, the repair heuristic used could incorporate continuous robot dynamics and thus take even longer than the repair heuristics used in our discrete setting; this would further reduce the relative model overhead.
> 2. Both MAPF-ML-LNS and our work requires multiple calls to the destroy heuristic. However, the destroy heuristic takes 3-4 orders of magnitude less time than the repair heuristic, and thus is insignificant in the overall runtime.
> 3. PBS is indeed slower than PP, but offers significantly stronger improvements per call, especially as we show in parameter sweeps in Appendix A.2. The slower runtime of PBS is actually synergistic with our neural network approach (see Bullet 1).
>
> Regarding the reviewer’s concerns about generalization, our generalization experiments are grounded in multi-robot warehousing applications, where it would be uncommon (and expensive!) to have large deviations in floor maps (especially size) or warehouse operating conditions (e.g. 5x fewer robots). Under such circumstances, we presume it would be common practice to train separate models rather than rely on generalization. On the other hand, obstacle locations, number of agents, and subset construction could change on a minute-by-minute or week-by-week basis, and thus we felt that this warranted an investigation into generalization. The results of our generalization analysis is summarized in Table 2 and are favorable: we found that models trained on both the random and empty floor maps could generalize better than Linear to different number of agents and/or different subset construction heuristics/sizes in the empty floor map.

---

> ### Author Response · Authors · 2023-11-15
> **Response Part 2/3**
>
> **Significance/Evaluation**: We acknowledge that our original submission lacks direct comparisons that invites concerns regarding fairness/significance, especially the relative metrics that we used and the lack of per-iteration performance comparisons. We thus provide additional results which addresses these concerns and allow direct comparisons with MAPF-LNS and MAPF-ML-LNS.
>
> Our Linear baseline also uses the best fixed neighborhood size and destroy heuristic. As we explain in detail in updated Appendix A.2, neither our work nor MAPF-ML-LNS is designed to choose between subsets with different sizes in a principled manner. E.g. a subset of size 50 would take significantly longer time to repair than a subset of size 5, but neither our work nor MAPF-ML-LNS considers the runtime of repairing the subsets. Thus, to keep the experimental setting controlled, we do not consider subsets with different size. We note that any future linear-model work proposing a principled way to consider subsets of different size is also applicable to our neural method. Moreover, when comparing absolute performance below, MAPF-ML-LNS does not outperform our Linear (and we could not find their code for comparison). Thus, we believe that our implementation of linear guided LNS is fair to previous work.
>
> Regarding absolute performance, this is a great point which we aim to convincingly address here. We compile the best “sums of delays” for each of our studied settings from MAPF-LNS and MAPF-ML-LNS into Appendix A.7. We show that except for the empty (32x32) setting studied by MAPF-LNS, our sums of delays are significantly lower than both previous works. Unclear to us, MAPF-ML-LNS often reports significantly higher sums of delays than the original MAPF-LNS. While compute resources may be different, we believe that our sums of delays are reasonable or better compared to MAPF-LNS and MAPF-ML-LNS. The better solution qualities that we see may be explained by the difference between using PBS vs PP as the subproblem solver, as illustrated in Appendix A.2. Unfortunately, we could not compare AUC directly with either work, as they do not report absolute AUCs. We believe that our sums of delays are worse in the empty (32x32) setting than MAPF-LNS because this is the only setting where MAPF-LNS finds it advantageous to use EECBS [1] as the initialization algorithm (see MAPF-LNS Table 2), while we only use PP and PPS as the initialization algorithms.
>
> [1] Li, Jiaoyang, Wheeler Ruml, and Sven Koenig. "Eecbs: A bounded-suboptimal search for multi-agent path finding." Proceedings of the AAAI Conference on Artificial Intelligence. Vol. 35. No. 14. 2021.
>
> Regarding generalization performance to different number of agents, in Table 2 we show that our models are reasonably generalizable, as demonstrated between different number of training ($|A| = 350$ or $|A| = 250$) and testing agents ($|A| \in \\{250, 300, 350\\}$).
>
> Regarding reporting performance vs number of iterations, we indeed show that our method uses significantly fewer LNS iterations than baselines to attain a given sum of delays. In new Appendix A.8, we graph the sum of delays vs iteration for all settings. We find that the conclusions here are similar to Appendix A.7, and that our method reduces both the number of LNS steps and the runtime needed to attain a given solution quality.
>
> Regarding the reviewer’s request for code, we hope that the concern for reproducibility is alleviated by our additional analysis: 1) the absolute sums of delays comparisons with MAPF-LNS and MAPF-ML-LNS and 2) the per-iteration comparisons of learned vs unguided selection. We are committed to releasing full code and trained models for reproducibility upon publication. We are averse to releasing code before then, especially because the public nature of ICLR review process leaves some room for abuse of released material if our paper were not accepted. Please let us know if the reviewer has further concerns regarding this point, and we will be happy to see how we can address the concerns.

---

> ### Author Response · Authors · 2023-11-15
> **Response Part 3/3**
>
> **Questions**
>
> We study PBS as a repair heuristics because it is one of the state-of-the-practice solvers which is more scalable than CBS and offers much better solution qualities than PP. We are unclear on why MAPF-LNS and MAPF-ML-LNS do not study PBS, even though it is used in other papers by the same authors [2]. Thus, as MAPF-LNS only sweeps for PP from $k = 2$ to $k = 16$ and differs in compute resources from us, we must sweep hyperparameters ourselves with both PBS and PP. Indeed, our hyperparameter search finds that larger neighborhood sizes (e.g. $k = 25$) tend to be the best for PBS.
>
> [2] Li, Jiaoyang, et al. "Lifelong multi-agent path finding in large-scale warehouses." Proceedings of the AAAI Conference on Artificial Intelligence. Vol. 35. No. 13. 2021.
>
> In general, it also could make sense to use deep learning as the destroy heuristic for LNS and this could be explored by future work in the MAPF domain. [3, 4, 5] use deep learning as the destroy heuristics. On the other hand, [MAPF-ML-LNS, 6] are similar to ours and perform subset selection. In general, we find that deep learning works in (mixed) integer (linear) programming like [3, 4] tend to learn the destroy heuristic.
>
> [3] Y. Wu et al., "Learning Large Neighborhood Search Policy for Integer Programming", NeurIPS 2021
>
> [4] Huang, Taoan, et al. "Searching large neighborhoods for integer linear programs with contrastive learning." International Conference on Machine Learning. PMLR, 2023.
>
> [5] Zong, Zefang, et al. "Rbg: Hierarchically solving large-scale routing problems in logistic systems via reinforcement learning." Proceedings of the 28th ACM SIGKDD Conference on Knowledge Discovery and Data Mining. 2022.
>
> [6] Li, Sirui, Zhongxia Yan, and Cathy Wu. "Learning to delegate for large-scale vehicle routing." Advances in Neural Information Processing Systems 34 (2021): 26198-26211.
>
> We again thank the reviewer for taking the time to help us strengthen our work, and we truly believe that the reviewer’s suggestions has led to a more grounded perspective of our work. We hope that the reviewer will consider increasing our score accordingly if we have appropriately alleviated the reviewer's concerns.

---

> ### Comment · Reviewer_TyGL · 2023-11-21
> **Follow-Up**
>
> Thank you for the rebuttal (and apologies for the time it took me to read through it). Most concerns are addressed, and I will slightly raise my score.
>
> However, I do not acknowledge the lack of code. Since reproducibility and implementation details (especially regarding MAPF/search algorithms) are important in our community, the concerns regarding abuse should be outweighed by an adequate review since any code may have flaws. Thus, allowing independent checks can improve the overall quality of the work.

---

> > ### Author Response · Authors · 2023-11-21
> > **Link to and descriptions of MAPF/search code**
> >
> > Thank you for your response! Regarding the reviewer’s remaining code-level concerns for MAPF/search algorithms, we provide an anonymized repository https://github.com/hdfffdfccf/hdfffdfccf to our low-level MAPF implementations in C++ for all reviewers to inspect. In our implementation, we aim to provide an efficient and clean implementation of MAPF algorithms for future researchers. Although we focus on the learning aspect of our work in our paper, our PBS-based LNS may be of independent interest to practitioners. The rest of our software architecture is described in Section A.5 and also will support future ML-based research for MAPF in general. As mentioned previously, we are committed to reproducible science and will release full code upon publication.
> >
> > The key classes are (defined in their respective .h and .cpp files):
> >
> > 1. `LNS` (subclass of `Solver`): containing logic for destroy heuristics and calls to the given repair heuristics. In the same file, we also define
> >     1. `LNSConfig`: parameters for LNS
> >     2. `Subset`: a problem subset
> > 2. `PBS` (subclass of `Solver`): PBS as a repair heuristic
> >     1. `PriorityGraph`: used to define the agent priority graph during PBS
> > 3. `PP` (subclass of `Solver`): PP as a repair heuristic, as well as for initialization
> > 4. `AStar` (subclass of `PathPlanner`): A* as single-agent path finding
> > 5. `ArrayOccupancy` (subclass of `Occupancy`): defines the spatiotemporal occupancy
> > 6. `UndirectedGraph` (subclass of `MapGraph`): defines the floor map
> > 7. common.h/cpp defines several shared data structures
> >     1. `Config`: run config
> >     2. `State`: a location and timestep of an agent
> >     3. `Path`: vector of `State`s
> >     4. `Problem`: vector of start states and goal locations. While we do not use multiple goals for an agent for this work, our `Problem` class handles this.
> >     5. `Solution`: vector of paths for each agent, the corresponding costs, and runtimes/number of search iterations
> >
> > We make several notes clarifying our code here:
> >
> > 1. We incorporated elements from two codebases [1,2]:
> >     1. We refactored the PBS implementation from https://github.com/Jiaoyang-Li/RHCR/blob/master/inc/PBS.h [1] to adapt it to the stay-at-goal MAPF formulation. Agents in [1] disappear at the goal, unlike in the more standard MAPF-LNS [2].
> >     2. Our ArrayOccupancy is refactored from https://github.com/Jiaoyang-Li/MAPF-LNS/blob/master/inc/PathTable.h [2]. We found that array-based occupancy was significantly faster than the interval-based occupancy used in https://github.com/Jiaoyang-Li/RHCR/blob/master/inc/ReservationTable.h [1]. Therefore, we only use array-based occupancy in our code.
> >     3. Our Prioritized Planning (PP) is refactored from https://github.com/Jiaoyang-Li/MAPF-LNS.
> >     4. Like [2], we also used A* as the single-agent path planner. Our single-agent A* implementations are refactored considering both [1] and [2] to incorporate the stay-at-goal MAPF formulation.
> >     5. We removed many unused components / if-statements of previous codebases for clarity.
> > 2. For all C++ operations (e.g. calls to the PBS or PP solvers), we perform runtime measurement in C++. This eliminates communication/interfacing delays between C++ and Python (though it shouldn’t be significant anyways). We envision that any solution for an industrial system would be fully in C++ and not need to communicate with Python, so our benchmarking reflects this.
> > 3. We avoid copying of data structures whenever possible.
> > 4. Our initialization precomputes the `heuristics` table in `MapGraph` to enable fast computation of heuristic distances in the A* search.
> > 5. A few data structures, e.g. `IntervalListOccupancy`, are not used at all in our ICLR submission.
> >
> > [1] Li, Jiaoyang, et al. "Lifelong multi-agent path finding in large-scale warehouses." Proceedings of the AAAI Conference on Artificial Intelligence. Vol. 35. No. 13. 2021.
> >
> > [2] Li, Jiaoyang, et al. "Anytime multi-agent path finding via large neighborhood search." International Joint Conference on Artificial Intelligence 2021. Association for the Advancement of Artificial Intelligence (AAAI), 2021.
> >
> > [3] Phillips, Mike, and Maxim Likhachev. "Sipp: Safe interval path planning for dynamic environments." 2011 IEEE international conference on robotics and automation. IEEE, 2011.
> >
> > Overall, we tried our best to make efficient low-level design choices and to pay attention to details, and we are confident in our code’s quality. We appreciate the reviewer’s time in reviewing our code, and we would appreciate any further suggestions from the reviewer. If we have further addressed the reviewer’s concerns, we hope that the reviewer would consider further raising our score.

---

> > > ### Comment · Reviewer_TyGL · 2023-11-23
> > > **Follow-Up**
> > >
> > > Thank you for the code! I appreciate the easy readability of the functions.
> > >
> > > I tried to follow the code logic, but I could not find a concrete implementation of the virtual method ``solve`` in `LNS.h` or `LNS.cpp`. Is that the high-level solver with the machine learning logic that has been left out (since I cannot find any plug-in points to a learned model)?
> > >
> > > How am I supposed to validate the main contribution of the paper, when it is not included?

---

> > > > ### Author Response · Authors · 2023-11-23
> > > >
> > > > Thank you for taking a look at our code!
> > > >
> > > > Yes, the high level logic is instead illustrated in Figure 1. As the reviewer noted in their earlier response, the MAPF/search algorithm implementations are more important to review and thus we have provided exactly these low-level MAPF components: the destroy heuristics / subset construction methods and the repair heuristics / PBS / PP. Since these represent the bulk of the search-based logic, we believe that reviewing these constitutes more than an adequate review of the code, considering the detailed comparisons provided by our empirical results. Moreover, we are more willing to release these low-level code because similar code can be found in other repositories already.
> > > >
> > > > The high-level logic and interface with our low-level code has been described by our paper in much detail already, and we do not believe that the value of verifying every line of our source code outweighs the risks of abuse. Releasing the full code could allow others to directly utilize our work for their own gains if it were not accepted by ICLR, and the high-level neural code is where the majority of our novelty lies. In our experience, it is not typical for ICLR submissions to provide code before publication, and we feel that the low-level code that we have released is more than adequate to verify the soundness of our implementation, given our empirical results and baseline comparisons. If there were a bug in the neural network, then it would be impossible for us to obtain the empirical results reported in our Table 1 and Appendix 7 and 8. We hope that the reviewer understands our predicament as authors who have dedicated much time and effort to our source code.

---

> > > > > ### Comment · Reviewer_TyGL · 2023-11-23
> > > > > **Follow-Up**
> > > > >
> > > > > I see. Given that the machine learning community is shifting towards better reproducibility and transparency, it is sad that code still needs to be concealed like this. It is difficult for me to understand why revealing code is such a big issue in this particular case.
> > > > >
> > > > > Thus, I am sorry that there is nothing more that I can do at this point.

---

### Official Review · Reviewer_TxMF · 2023-10-30

**Soundness:** 3 good
**Presentation:** 3 good
**Contribution:** 3 good
**Rating:** 6
**Confidence:** 4

**Summary:**

This paper is a study that utilizes a neural network structure based on 3D convolution to perform multi-agent path finding from a spatial-temporal perspective. Unlike previous research that used linear feature-based machine learning structures, this paper eliminates the feature dependency through 3D convolution-based architecture.

By selecting a subset of k diverse combinations of agents from the entire agent set and comparing the changes in cost, it distinguishes between the multi-subset approach, which aims to find the optimal subset structure, and the per-subset approach, which uses only a single subset. Through testing, it confirms that the multi-subset approach performs better. Furthermore, it also demonstrates performance improvements when comparing with the Unguided method, which extracts subsets using predefined rules, and the traditional Linear method.

**Strengths:**

By utilizing 3D convolution to consider spatial and temporal information simultaneously, the authors have proposed a method for enabling appropriate path finding for multi-agents without collision in situations involving a large number of agents at higher speed. Furthermore, through the Multi-subset structure, they solve multiple subproblems in a "batch" format, which has the advantage of quickly verifying better paths using this batch structure.

**Weaknesses:**

The key point of this paper is the rapid resolution of Guiding LNS using 3D convolution without the need for separate feature design in the network structure. However, the author has only applied 3D convolution without providing further theoretical proof or proposing new methods. Therefore, while it is acknowledged that there is an improvement in performance through the application of deep learning structures, the paper has limitations in terms of its overall value.

Additionally, the MAPF the author aimed to address only holds significance when applied to real robots. However, the author tested the proposed method in a simplified simulation environment for performance verification. This aspect restricts the paper's value to the theoretical domain. To give this research more meaning, it would have been beneficial to include tests involving the use of robots in real-world environments, such as logistics robots.

**Questions:**

Overall, the representation through diagrams is lacking. First, this paper places significant importance on the Time dimension. However, it merely mentions the T dimension without providing any illustrative examples of path changes in this time domain direction, which made it challenging to comprehend. It would have been easier to understand if a few examples of images with different appearances in the T dimension were shown.

Additionally, it would have been helpful for understanding if the paper had diagrammatically represented the network in Figures 2 and 3. The section regarding the network is written with text such as "3D CNN" and "2D CNN," making it difficult to intuitively grasp. Visualizing the network, as done in other papers that use convolution layers, would have aided in conveying the content.

Furthermore, in Section 4.3, two types of transformers are utilized: Light-weight and Heavy-weight. It would be beneficial if the differences between these two structures were more explicitly mentioned.

**Details Of Ethics Concerns:**

No concern.

---

> ### Author Response · Authors · 2023-11-15
>
> We very much thank the reviewer for alerting us to the deficiencies of our current architectural Figures 2 and 3, which we have completely revamped during the rebuttal. We would appreciate the reviewer taking some time to examine our new Figures 2 and 3, which can be viewed by re-downloading the paper from OpenReview. Furthermore, we encourage the reviewer to view our general response to all reviewers, where we also discuss other major rebuttal additions. Here we address some of the reviewer’s comments.
>
> **Weaknesses**
>
> 1. We acknowledge that our work does not aim for theoretical impact; however, we believe that our work provides a new and valuable empirical method. The integration of intra-path attention with 3D convolution is the core aspect of our neural method, and is broadly applicable to general multi-path problems, which occur in multi-agent or combinatorial problems that consider spatiotemporal interactions. Though we focus on LNS, we envision that the same architecture may improve the performance on classification, regression, generative, and decision tasks in these domains.
> 2. While we do not consider real-world robot experiments in this work, it is a great direction for future work (we have added it to the conclusions). Even without such experiments, our work informs practitioners (e.g., logistics companies) of the potential and limitations of deep neural approaches for LNS-based methods.
>
> **Questions**
>
> 1. We have significantly updated our Figures 2 and 3 to better illustrate the time dimensions. We would appreciate any additional feedback on the new figures!
> 2. Similarly, please let us know if the updated Figures 2 and 3 allow readers to better visualize the convolutional structures.
> 3. In our updated Figure 3, we have better illustrated “light-weight” vs “heavy-weight”. The “heavy-weight” transformer attention operations attends across each agent’s entire path (and is thus proportional to the number of agents and T); these operations are contained within the “Convolution-attention block” and are shared across all subsets. The “light-weight” attention operation only operates on the **first** tensor along each subset agent’s path rather than the entire path, and thus does not depend on T. Thus, both “heavy-weight” and “light-weight” attention utilize the standard transformer architecture, but differ in the size of the inputs, and thus their computational cost.
>
> By following the reviewer's suggestions, we believe we have significantly strengthened our submission. If we have addressed the reviewer's concerns, we hope that the reviewer will consider increasing their score accordingly.

---

> > ### Comment · Reviewer_TxMF · 2023-11-21
> >
> > Thank you for the rebuttal. Firstly, I appreciate the clear representation of the Time dimension in Figures 2 and 3, which was helpful for understanding. However, it seems necessary to maintain a consistent arrangement of columns in the figures. They are arranged irregularlly now. Additionally, the previous use of opacity to depict paths in the figures was helpful for comprehension, but it seems to be missing in the current illustration, leading to a decrease in clarity. Due to the absence of paths and the representation of only nodes, it is challenging to distinguish between the current path and the shortest path. If the paths were removed due to complexity, reducing the number of layers in the Time dimension direction and using opacity for each layer to represent the entire path might enhance understanding.
> >
> > Additionally, in Figure 3, the proximity between the upper and lower rows causes confusion. It would be beneficial to add a slight margin between them. Moreover, the figure is too small, and the text is not easily visible unless enlarged. Eliminating the left and right margins and increasing the figure size would likely improve visibility.

---

> > > ### Author Response · Authors · 2023-11-22
> > >
> > > Thank you for your close read of our new figures and your detailed additional pointers! We have updated our figures accordingly:
> > >
> > > 1. We have added an additional component to Figure 2, illustrating the agent paths and agent shortest paths, and referenced this component in the caption. We tried using opacity here initially, but found that there was too much overlap between paths of different agents, and that arrows may better illustrate the agent paths than opacity. We have referenced these paths in the caption of Figure 3 as well.
> > > 2. We have increased the gap between the upper and lower rows in Figure 3. We have increased the text sizes for both Figures 2 and 3 and eliminated the margins.
> > > 3. We noticed that we exported to png previously, resulting in slightly blurry figures. We have fixed this now and the figures should be sharp.
> > >
> > >
> > > > It seems necessary to maintain a consistent arrangement of columns in the figures. They are arranged irregularly now.
> > >
> > > We are unsure what the reviewer means here by "columns" and "irregularity". If the "irregularity" refers to the varying dimensions of the obstacles, position, and time embeddings, then we would like to clarify that we are performing element-wise additions broadcasted along the temporal dimension for the obstacle and position embeddings and broadcasted along the two spatial dimensions for the time embedding.
> > >
> > > We hope that we have further addressed the reviewer’s concerns with these modification. If these changes and our overall rebuttal changes have indeed strengthened our paper, we hope that the reviewer could consider raising our score.

---

### Official Review · Reviewer_CV86 · 2023-11-01

**Soundness:** 3 good
**Presentation:** 3 good
**Contribution:** 3 good
**Rating:** 6
**Confidence:** 5

**Summary:**

1The paper proposes to use a deep learning-based framework to select agent subsets to destroy in LNS for the problem of multi-agent path finding (MAPF). Two architectures are proposed: per-subset and multi-subset methods. In experiment, two architectures are tested on five maps of different sizes. In particular, multi-subset performs a lot better than the other approaches.

**Strengths:**

1. The deep neural network architecture that incorporates spatiotemporal information seems to be a good contribution to the MAPF community.

2. Empirical results demonstrate the usefulness of the multi-subset architecture.

**Weaknesses:**

1. The description of the runtime overhead is a bit unclear. I just want to confirm since this is important: when you run your method for 60 seconds, this includes the machine learning inference overhead, right? Though, from the plot it is clear your approach is still the best when the overhead is included.

2. Have you tried generalizing your model to other agent sizes on the same map as training (similar to what Huang et al show in their paper)?

3. It seems unfair to restrict the unguided approach to use only one destroy heuristic, this restriction is made mainly for the benefit of the ML-guided approaches. In the MAPF-LNS paper, it has been shown that the adaptive destroy heuristic is much better for the unguided approach.

**Questions:**

You mentioned two challenges in the intro with the second one being the overhead of machine learning inference time. Can you elaborate more on how you address this issue with your model?
There is a short paragraph in section 5 that describes some of the engineering details. Do you use other techniques behind the scenes?


You use c_min to compute the gap, which is the solution found by the baseline within 10 minutes. Could this lead to a negative gap when your method finds better solutions within just 1 minute? (though it is unlikely to happen)

---

> ### Author Response · Authors · 2023-11-15
>
> We thank the reviewer for the feedback and suggestions. Here are our responses to each concern:
>
> **Weaknesses**
>
> 1. Yes, the Average Gap (%), Win / Loss, and Final Gap (%) in Table 1 already include the stated model overhead in the 60s runtime (they are exactly the solid curves in Figure 5).
> 2. Yes, in Table 2, we did indeed find that our method generalizes better than Linear to different warehouse obstacle layouts (random (32x32) during training and empty (32x32) during testing), different destroy heuristics (agent-local $k=25$ during training and uniform $k=50$ during testing), and different number of agents ($|A| = 250$ or $|A| = 350$ during training and $|A| \in \\{250, 300, 350\\}$ during testing). Regarding how we chose these generalization configurations: our generalization experiments are grounded in multi-robot warehousing applications, where it would be uncommon (and expensive!) to have large deviations in floor maps (especially size) or warehouse operating conditions (e.g. 5x fewer robots). Under such circumstances, we presume it would be common practice to train separate models rather than rely on generalization. On the other hand, obstacle locations, number of agents, and subset construction could change on a minute-by-minute or week-by-week basis, and thus we felt that this warranted an investigation into generalization.
> 3. Adaptive LNS (ALNS), which is used by MAPF-LNS and MAPF-ML-LNS to adaptively adjust subset construction method (destroy heuristic), does not offer significant difference in performance compared to the **best** fixed subset construction method used in our paper. Indeed, examining Table 2 in the original MAPF-LNS paper, ALNS is often worse than the best fixed destroy heuristic. We also demonstrate this in our sweep experiments in Appendix A.2, Table 7 and Table 6. Therefore, we believe that it is fair to restrict our focus to the best fixed destroy heuristics.
>
> **Questions**
>
> 1. As suggested by reviewers, we revamped illustrations of our architectures in updated Figures 2 and 3 in the paper (please see the updated pdf on OpenReview). Comparing Figure 2 and Figure 3, the Multi-subset architecture shares much of the computation across all $J$ subsets, and the computation specific to each subset is relatively minimal and requires no 3D convolution or attention across space and time. As discussed in Section 5, we may use spatiotemporal pooling for all agent locations and obstacles for both Multi-subset and Per-subset. We also use fp16 precision instead of fp32 precision. Other than these techniques already stated in our paper, we have not used additional techniques behind the scene, though we believe that the model overhead can be further improved if needed with traditional model compression techniques like distillation, pruning, and/or quantization.
> 2. Indeed, this could lead to the gap turning negative, but that would require a more than 10x speedup over the Unguided LNS baseline.
>
> We hope that we have addressed the reviewer’s concerns here, and we encourage the reviewer to additionally take into account new results presented in our general rebuttal to all reviewers. We hope that the reviewer will consider increasing our score if they agree that we have strengthened the work.

---

> > ### Comment · Reviewer_CV86 · 2023-11-23
> >
> > I have read the rebuttal and read the other discussions. I trust the results but strongly encourage the authors to fully open-source the code should it get accepted.

---

> > > ### Author Response · Authors · 2023-11-23
> > >
> > > We thank the reviewer for their time in looking over our rebuttal and following the various discussions! We commit to fully open sourcing our code, and indeed we believe that our Python/C++ interface would be particular impactful in aiding future research in learning-based MAPF. Should our work be accepted, we look forward to the opportunity to providing these tools to the community!

---

### Author Response · Authors · 2023-11-15
**General Response to all Reviewers**

We are grateful to all reviewers for their time and actionable suggestions. We took these to heart and addressed all points experimentally or otherwise providing clarifications and reasoning. In this general response, we discuss central topics relevant to multiple or all reviewers. While we summarize our additional findings on OpenReview, we encourage reviewers to download the updated paper pdf from OpenReview again, especially because we have included new figures which are difficult to convey in the rebuttal as text and added new Appendix sections (highlighted in red for visibility).

**Clarifying the key contribution**: Thanks to overall reviewer feedback, we revised our descriptions of the key contribution of our work - the proposed Multi-Subset architecture. Multiple reviewers (TxMF and TyGL) interpreted our initial submission as conducting a straightforward application of common architectures (3D CNN and Transformer attention). In the updated paper, we provide greater clarity on the design principles behind the architecture and the resulting intra-path attention mechanism. This is illustrated with our updated Figure 3. In short, the design principles are 1) to permit *trajectory-level* information to flow between any two agents whose paths intersect, and 2) amortize computation across multiple subproblems. While 3D convolution capture local agent-agent and agent-obstacle interactions, our designed intra-path attention allows information to additionally propagate along an agent’s entire path and capture long-range path-level interactions. The result is a representation in which any two agents are two-hop neighbors if their paths both interact some common agent at any point. As the representation can be disaggregated into individual agent trajectories, we can regroup individual agent representations efficiently into arbitrary number of subsets as required by LNS. Additionally, our use of Transformer self-attention is non-standard; in particular, our architecture attends between tensors along agents’ paths extracted from a 3D spatiotemporal tensor through a series of gather and scatter operations. In support of the new architecture, we have also included a new Appendix A.6, with a performance ablation for the impacts of the 3D convolution and intra-path attention mechanisms; we show that the intra-path attention especially is critical for the performance of our architecture.

**Direct comparison to prior work**: Reviewer TyGL encouraged us to offer absolute comparisons in final solution qualities reported by unguided and linear-guided previous works: MAPF-LNS [1] and MAPF-ML-LNS [2]. We have now added direct comparisons to these works in Appendix A.7, and we show that our baseline Unguided generally obtains better performances than attained by those previous works, in part due to our use of PBS [3] as a subproblem solver rather than PP.

**Inclusion of neural network overhead**: Addressing reviewers CV86 and pDBe regarding whether we account for overhead of our neural network, the main results table (Table 1) indeed includes model overhead towards the 60s runtime when calculating our Average Gap (%), Win / Loss, and Final Gap (%).

**Performance vs LNS iteration**: Prompted by Reviewer TyGL, we have added Appendix A.8 demonstrating that the Multi-Subset architecture requires significantly fewer **LNS iterations** to attain any given solution qualities. These new experiments further affirm that the reduction in number of LNS iterations by our Multi-Subset network (to attain a given solution quality) substantially outweighs the additional overhead due to the neural network.

We very much appreciate the suggestions of all reviewers, as we truly believe that the suggestions significantly strengthen our work. Below, we respond to each reviewer’s remaining feedback individually.

[1] Li, Jiaoyang, et al. "Anytime multi-agent path finding via large neighborhood search." International Joint Conference on Artificial Intelligence 2021. Association for the Advancement of Artificial Intelligence (AAAI), 2021.

[2] Huang, Taoan, et al. "Anytime multi-agent path finding via machine learning-guided large neighborhood search." Proceedings of the AAAI Conference on Artificial Intelligence. Vol. 36. No. 9. 2022.

[3] Ma, Hang, et al. "Searching with consistent prioritization for multi-agent path finding." Proceedings of the AAAI conference on artificial intelligence. Vol. 33. No. 01. 2019.

---

### Public Comment · ~Yudong_Luo1 · 2024-04-26
**Questions on the experiment setting**

Dear authors, after reading the paper, I find the experiment setting is not clearly described.

+ For dataset collection: how many scenes are used for a map? "we execute LNS for 25 to 100 improvement iterations" how to determine this iteration number; "J=100" does this mean at each iteration, we propose 100 neighorhood sets? and which heuristic is used to select neighborhood?

+ For model training, based on Sec 6.3, it seems separate models are trained for separate maps, I would like to confirm if my understanding is correct.

+ For evaluation, in each iteration, does the algorithm still propose 100 neighborhood candidate sets as in training? Is this time included in the total time?

Thank you.

---

> ### Public Comment · ~Zhongxia_Yan1 · 2024-04-26
>
> Hi Yudong,
>
> Thanks for your interest!
> 1. 25 to 100 was somewhat arbitrarily chosen, since data collection is the most computationally expensive and we did not experiment extensively with the amount of data collected. Essentially we ran enumeration until the solution cost nearly plateaus. We will release all experimental parameters soon, and you will find the individual settings there. Yes, J is the number of proposed subsets at each iteration. The heuristic and subset sizes were chosen in the sweeps in Appendix A.2, and listed in Table 1 (i.e. "Agent-local k = 25" corresponds to the agent-local heuristics described in MAPF-LNS [Li 2021].
> 2. Yes, this is correct except for in Table 2, where we tried to transfer between maps.
> 3. Yes, evaluation still proposes 100 subsets/iteration, though we did not experiment with this parameter much. As mentioned to Reviewer TyGL earlier, we did not really consider the subset proposal time as 1) all methods except Unguided use the same subset proposal heuristics, 2) the destroy heuristics take 3-4 orders of magnitude less time than the subset solving times.
>
> We've already pushed the code online, but are still working on pushing run configs and model checkpoints!

---

> > ### Public Comment · ~Yudong_Luo1 · 2024-05-01
> > **Thank you for the reply.**
> >
> > Thank you for the prompt and detailed reply!

---

### Meta-Review · Area_Chair_7fNa · 2023-12-10

**Metareview:**

This is a learning-to-plan paper that proposes a method to incorporate a learned heuristic in multi-agent path planning. The paper starts from an existing state-of-the-art method, MAPF-LNS (Multi-Agent Path Finding - Large Neighbourhood Search), which decomposes a large planning problem into a sequence of smaller sub-problems. MAPF-LNS starts with a feasible solution and then iteratively selects subsets of paths to improve, while the unselected path remain fixed. Learning-to-plan methods have addressed this problem before, for example, MAPF-ML-LNS, which uses a learned linear model with handcrafted features to prioritize which group to select. The paper herein proposes an improvement: it introduces a 3D convolution architecture to learn scores that will help prioritize these groups in the same order that the (slow) ground truth solver would. The paper shows consistent improvements over MAPF-LNS, MAPF-ML-LNS and other baselines, and thus, I think it is worth acceptance at the conference.

**Justification For Why Not Higher Score:**

The novelty of the paper does not warrant a spotlight presentation, given that this paper is an improvement over MAPF-ML-LNS.

**Justification For Why Not Lower Score:**

I don't think the paper should be rejected.

---

### Decision · Program_Chairs · 2024-01-16

Accept (poster)